# Regulation of protein abundance in genetically diverse mouse populations

## Graphical abstract

## Authors

Gregory R. Keele, Tian Zhang,
Duy T. Pham, ..., Martin T. Ferris,
Steven P. Gygi, Gary A. Churchill

## Correspondence

gary.churchill@jax.org

## In brief

Keele et al. report quantitative profiling of protein abundance in liver tissue from genetically diverse mouse populations, powerful resources for characterizing protein regulation. They map QTLs for individual proteins and complexes and find conservation of local genetic effects and sex differences for individual proteins across inbred and outbred mice.

## Highlights

- Quantitative profiling of liver proteomes of genetically diverse mouse populations

- Consistent genetic effects on individual proteins between inbred and outbred mice

- Proteins in complexes show reduced heritability compared with proteins not in complexes

- *De novo* mutations in inbred strains contribute to protein abundance variation

Keele et al., 2021, Cell Genomics 1, 100003
October 13, 2021 © 2021 The Author(s).

# Cell Genomics

CellPress

## Article

# Regulation of protein abundance in genetically diverse mouse populations

Gregory R. Keele,[1,5] Tian Zhang,[2,5] Duy T. Pham,[1] Matthew Vincent,[1] Timothy A. Bell,[3] Pablo Hock,[3] Ginger D. Shaw,[3] Joao A. Paulo,[2] Steven C. Munger,[1] Fernando Pardo-Manuel de Villena,[3,4] Martin T. Ferris,[3] Steven P. Gygi,[2] and Gary A. Churchill[1,6,*]

[1]The Jackson Laboratory, Bar Harbor, ME 04609, USA
[2]Harvard Medical School, Boston, MA 02115, USA
[3]Department of Genetics, University of North Carolina, Chapel Hill, NC 27599, USA
[4]Lineberger Comprehensive Cancer Center, University of North Carolina, Chapel Hill, NC 27599, USA
[5]These authors contributed equally
[6]Lead contact
*Correspondence: gary.churchill@jax.org

## SUMMARY

Genetically diverse mouse populations are powerful tools for characterizing the regulation of the proteome and its relationship to whole-organism phenotypes. We used mass spectrometry to profile and quantify the abundance of 6,798 proteins in liver tissue from mice of both sexes across 58 Collaborative Cross (CC) inbred strains. We previously collected liver proteomics data from the related Diversity Outbred (DO) mice and their founder strains. We show concordance across the proteomics datasets despite being generated from separate experiments, allowing comparative analysis. We map protein abundance quantitative trait loci (pQTLs), identifying 1,087 local and 285 distal in the CC mice and 1,706 local and 414 distal in the DO mice. We find that regulatory effects on individual proteins are conserved across the mouse populations, in particular for local genetic variation and sex differences. In comparison, proteins that form complexes are often co-regulated, displaying varying genetic architectures, and overall show lower heritability and map fewer pQTLs. We have made this resource publicly available to enable quantitative analyses of the regulation of the proteome.

## INTRODUCTION

Protein abundance in cells is regulated at multiple levels, including transcriptional and various post-transcriptional, translational, and protein degradation mechanisms.[3] Each of these regulatory mechanisms can be influenced by genetic variation, as observed across a range of organisms, including *Arabidopsis*,[4] yeast,[5,6] mice,[1,7–9] and humans.[10–14] Genetic effects on protein abundance can be broadly divided into two classes: local and distal. Local variation in the vicinity of the coding gene typically influences protein abundance by altering the rate of transcription or stability of the transcript.[15] Distal genetic variation at loci far from the coding gene typically influences later stages of regulation, often acting through a diffusible intermediate such as another protein. Other modes of regulation are possible, and the local versus distal distinction is a useful but imperfect indicator for distinguishing translational from post-translational regulation. The complexity of genetic regulation is compounded for proteins that form multi-unit complexes because stoichiometry can impose varying degrees of constraint.[1,16–19] In each genetic context, proteins can be influenced by a single locus (monogenic) or many (multigenic to polygenic), and effects can be additive or dominant and may involve epistatic interactions with other loci.

Resource populations with high levels of genetic diversity can be used to identify and characterize the genetic loci that affect protein abundance. The Collaborative Cross (CC)[20,21] and Diversity Outbred (DO)[22] mouse populations are two genetic resource populations that are descendant from a common set of eight inbred strains (i.e., the founder strains; short names in parentheses): A/J (AJ), C57BL/6J (B6), 129S1/SvImJ (129), NOD/ShiLtJ (NOD), NZO/HILtJ (NZO), CAST/EiJ (CAST), PWK/PhJ (PWK), and WSB/EiJ (WSB). The founder strains represent three subspecies of the house mouse, *Mus musculus*,[23,24] and encompass genetic variation from across laboratory and wild mice. Each DO mouse is genetically unique with high levels of heterozygosity and low linkage disequilibrium (LD) that support fine mapping of genetic variants.[25–28] CC mice consist of more than 60 inbred strains that are homozygous at most loci (>99%).[21,29,30] They have larger LD blocks and thus lower mapping resolution because of fewer outbreeding generations in their derivation than DO mice. The genomes of CC strains are inbred and thus replenishable, enabling repeated measurements of genetically identical mice[31,32] within and across experiments as well as characterization of strain-specific phenotypes.[33–35] CC strains can model human diseases; examples include colitis,[36] susceptibility to Ebola infection,[37] influenza A virus,[38]

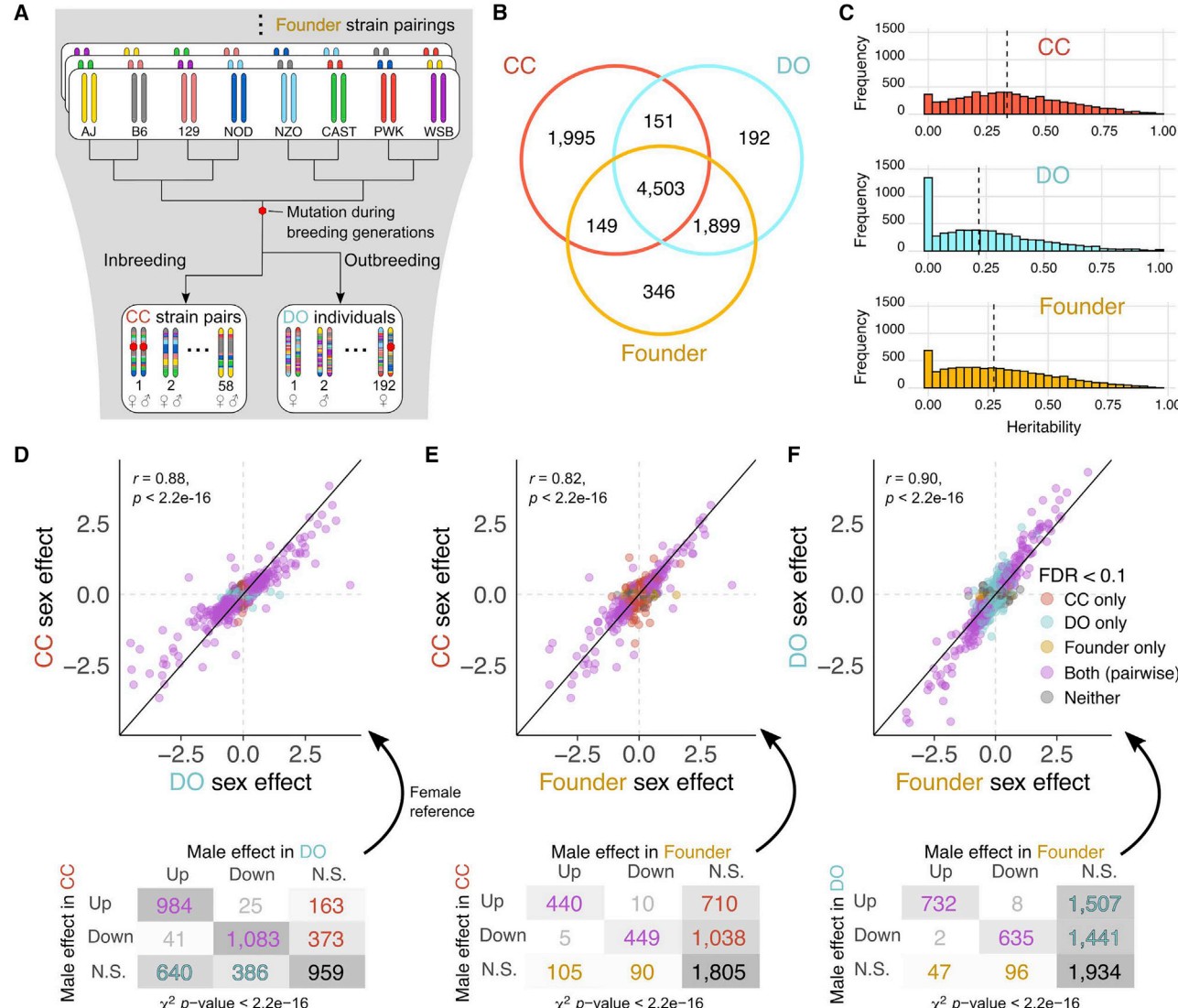

**Figure 1. Comparisons of genetic and sex effects on protein abundance among the CC, DO, and founder strains reveal strong concordance**

(A) The CC strains and DO mice are descended from the same eight inbred founder strains. Mutations occur during the breeding generations of the CC and DO mice and can become fixed in the CC strains.

(B) Venn diagram of the proteins analyzed in the CC, DO, and founder strains. The founder strains and DO samples were obtained in the same experiment, resulting in greater overlap.

(C) Estimates of heritability of protein abundance are greater on average in the inbred CC and founder strains compared with DO mice. Vertical lines represent the median heritability in each population.

(D–F) Sex effects for protein abundances in (D) CC versus DO, (E) CC versus founder strains, and (F) DO versus founder strains. The solid identity line and dashed horizontal and vertical lines at 0 are included for reference. Pearson correlation coefficients ($r$) between the sex effects of the populations and corresponding p values included. A breakdown of the direction of sex effects is shown for each comparison of populations. N.S. indicates proteins that did not have significant sex effects at FDR < 0.1. $\chi^2$ test of independence used to evaluate consistency of the direction of sex effects.

See also Figure S1 and Tables S1, S2, and S3.

severe acute respiratory syndrome (SARS) coronavirus,[39] and peanut allergy.[40]

In this study, we quantified protein abundance in liver samples of 116 CC mice representing female/male pairs from 58 strains (Table S1). We previously collected proteomics data from the livers of 192 DO mice and 32 mice representing the eight founder strains (two animals of each sex per founder strain)[1] (Figure 1A).

Both studies employed tandem mass tag (TMT) multiplexed mass spectrometry (MS) but represent separate experiments with differences that reflect refinements in the protocols (STAR Methods), most notably use of a pooled bridge sample in each TMT plex for the CC. We compare sex differences and quantitative trait loci for protein abundance (pQTLs) between CC and DO mice, finding strong conservation of sex differences and local

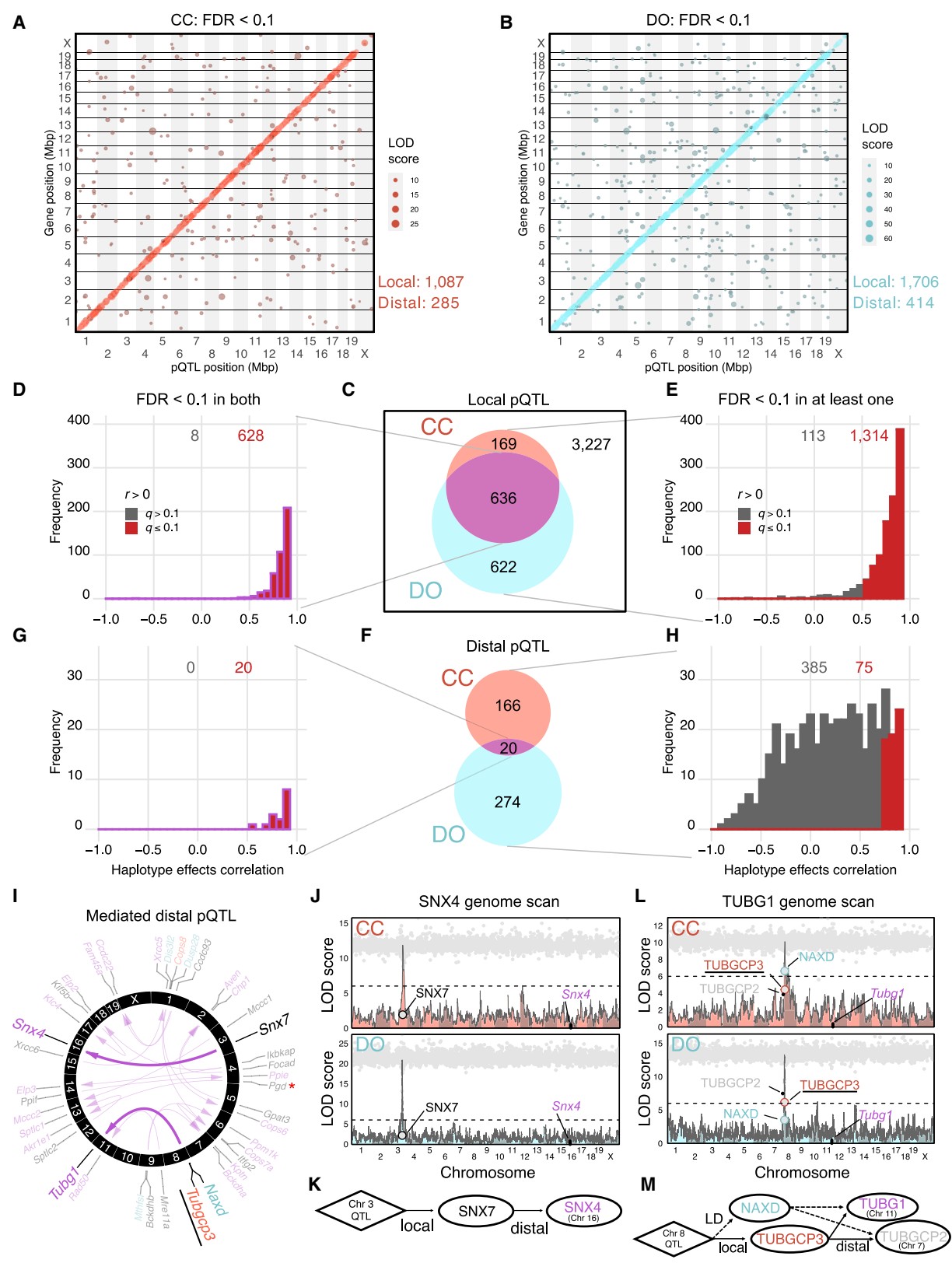

pQTLs and, to a lesser degree, distal pQTLs. We examine proteins that form complexes and find fewer local genetic effects. We highlight examples of protein complexes with diverse genetic architectures. We identify proteins showing unusual expression patterns in specific CC strains and associate some of these with *de novo* mutations in the CC strains and their phenotypic consequences. Our work demonstrates the consistency of MS proteomics data across experiments and conservation of the regulation of protein abundance across these resource populations.

## RESULTS

### Heritability and sex effects on proteins are shared across the CC, DO, and founder strains

We quantified the abundance of 6,798 proteins (Table S2) in liver tissue from 58 inbred CC strains, one female and one male per strain. We previously reported quantification of proteins from liver tissue of 192 outbred DO mice and 32 mice representing the eight founder strains (two per sex per strain).[1] The data for DO and founder strains were re-analyzed for this study to ensure that all data were processed consistently (STAR Methods), resulting in quantification of 6,745 and 6,897 proteins (Table S2), respectively. Among the 9,235 proteins detected in total, 4,503 were seen in all three populations (Figure 1B; Table S3).

We estimated protein abundance heritability ($h^2$), which reflects the combined effects of genetic factors relative to the precision of protein abundance estimation (Figure 1C). Heritability was higher on average in the CC and founder strains compared with the DO strain, likely because of the combined effects of their inbred genetic architecture and improved precision of the MS measurement for the CC samples. Despite the differences in average heritability, heritability of individual proteins was correlated significantly across populations (Pearson correlation coefficient [$r$] > 0.43, p < 2.2e−16), suggesting that the underlying genetic factors are conserved.

Protein abundance can differ between sexes.[1,18] We characterized sex effects in the CC, DO, and founder strains (STAR Methods; Table S3) and detected significant sex effects (false

discovery rate [FDR] < 0.1) for 3,750 (55.2%) proteins in the CC strains, 4,520 (67.0%) proteins in the DO mice, and 1,583 (23.0%) proteins in the founder strains. Sex-specific differences in protein abundance were overwhelmingly in the same direction for all populations (Figures 1D–1F). Gene set enrichment analysis revealed that proteins related to ribosome, translation, and protein transport Gene Ontology (GO) terms were more abundant in males,[18] whereas proteins related to catabolic and metabolic processes, including fatty acid metabolism, were more abundant in females in all populations.[41]

### Genetic regulation of proteins is shared between the CC and DO

To identify the genetic loci that regulate variation in protein abundance, we carried out pQTL mapping in CC and DO mice (Table S4). To determine significant pQTLs, we first applied a permutation analysis[42] to control the genome-wide error rate for each protein and then applied an FDR adjustment (FDR < 0.1) across proteins[43] to establish a stringent detection threshold for pQTLs. Using this stringent criterion, we identified 1,087 local and 285 distal pQTLs in CC mice and 1,706 local and 414 distal pQTLs in DO mice (Figures 2A and 2B). We defined local pQTLs as being located within 10 Mbp of the midpoint of the protein-coding gene. Although this wide local window may result in some nearby but distal pQTLs being misclassified as local, it accounts for the large LD blocks in the CC strains and yields more consistent classification of local pQTLs between CC and DO mice. We also identified a local pQTL on the mitochondrial genome in CC mice for *mt-Nd1* (Figure S1D). Stringent control of false positive rates can result in a high rate of false negative results. Therefore, to compare pQTL discovery across populations, we carried out a parallel analysis with more lenient FDR control (FDR < 0.5; Figures S1A and S1B).

We compared genetic effects between CC and DO mice by focusing on the 4,654 proteins that were detected in both populations (Figure S1C). Among 1,427 local pQTLs detected in either population, 636 were detected in both (Figure 2C). To determine whether the shared local pQTLs were driven by the same genetic variants, we compared the estimated haplotype effects at each

**Figure 2. Genetic effects of loci are highly consistent between CC and DO mice**

(A and B) Stringently detected pQTLs (FDR < 0.1) in (A) CC and (B) DO mice. The pQTLs are plotted by the genomic positions of protein-coding genes against pQTL location. Dot size is proportional to strength of association (log-odds [LOD] score).

(C) Venn diagram of local pQTLs detected in CC and DO mice.

(D) The correlation of haplotype effects for local pQTLs detected in CC and DO mice.

(E) The correlation of haplotype effects for local pQTLs detected in at least one of the CC or DO populations. Red bars represent pQTLs with significantly correlated effects (FDR < 0.1).

(F) Venn diagram of distal pQTLs detected in CC and DO mice.

(G) The correlation of haplotype effects for distal pQTLs detected in CC and DO mice.

(H) The correlation of haplotype effects for distal pQTLs detected in at least one of the CC or DO populations.

(I) The 20 distal pQTLs detected in CC and DO mice. Arrows connect candidate drivers identified through mediation analysis to their targets (proteins with distal pQTLs). Gene names in black represent the top candidates identified in both CC and DO mice. Red and blue gene names indicate top candidates specific to CC or DO mice, respectively. TUBGCP3, a top candidate from the CC and the stronger biological candidate based on shared membership in protein families, is underlined. The red asterisk denotes PGD as a likely false positive mediator because of the true mediator being unobserved, seen in both CC and DO mice.

(J) Mediation analysis of the *Snx4* distal pQTL, an example of agreement between CC and DO mice. Panels showing pQTL LOD scores for CC (pink) and DO (blue) mice are overlayed with mediation conditional LOD scores (gray dots). Mediation scores were evaluated for all proteins genome-wide. Candidate mediators of interest with low mediation scores are labeled. Horizontal lines at a LOD score of 6 are included as a reference point across genome scans.

(K) Causal diagram consistent with the relationships revealed by QTL and mediation analysis for SNX4 and SNX7.

(L) Mediation analysis of the *Tubg1* distal pQTL, an example of disagreement between CC and DO mice, details as described above.

(M) Causal diagram consistent with the relationships revealed by QTL and mediation analysis for TUBG1, TUBGCP3, TUBGCP2, and NAXD.

See also Figures S1–S3 and Tables S4 and S5.

pQTL (STAR Methods) and found that 628 (98.7%) were significantly positively correlated (FDR < 0.1; Figure 2D; Table S4). To assess whether pQTLs detected in only one population are population specific, we compared the haplotype effects of detected pQTLs with effects estimated at the corresponding locus in the other population regardless of significance and found that 1,314 (92.1%) were significantly positively correlated (FDR < 0.1; Figure 2E). The concordance of local pQTLs also holds for lenient detection (Figures S1F–S1G; Table S4). Based on these analyses, we find that local genetic effects on proteins are highly conserved between the CC and DO populations.

The founder strains can provide additional support for local pQTLs in CC and DO mice (Figures S1K and S2), particularly for those that are hard to detect because of rare alleles in CC or DO mice (e.g., observed in three or fewer CC strains). We selected all genes with a rare local founder haplotype that did not have a leniently detected pQTL in CC mice, representing 2,439 genes. We correlated the haplotype effects estimated at the locus closest to the gene transcription start sites (TSSs) with the protein abundance in the founders (STAR Methods) and found significant positive correlation for 194 genes (FDR < 0.1). The three populations together provide evidence of local genetic effects at 2,905 proteins.

Of the 186 distal pQTLs detected in CC mice and 294 in DO mice, 20 were detected in both populations, all with significantly correlated haplotype effects (FDR < 0.1; Figure 2G; Table S4). Overall, the distal pQTLs are weaker than the local pQTLs,[1,44] which may contribute to an increased rate of false negatives. By comparing haplotype effects for pQTLs that were detected in only one population, we identified an additional 55 shared distal pQTLs (Figure 2H). Based on the stringent criteria, a total of 75 distal pQTLs had consistent effects of the total (16.3%) detected in CC or DO mice compared with 19 (0.5%) for lenient criteria (Figure S1J). The reduction in concordance for the lenient criteria is likely due to increased numbers of false positives and weak distal pQTLs, although it did uniquely identify a shared distal pQTL for *Ercc3* (Figure S3).

## Mediators of strong distal genetic effects detected in CC and DO mice are concordant

The genetic variants that drive distal pQTL are generally thought to act through diffusible intermediates that are under local genetic control at the pQTL. We used mediation analysis[1,45–47] to identify candidate mediators of distal pQTLs (STAR Methods; Table S5). Mediation analysis is best used for prioritizing candidate mediators and, in this study, is limited to evaluating candidates that exert their effects through changes in protein abundance. Therefore, we cannot exclude the possibility of mediation by proteins that were not detected, by non-coding RNAs or by protein variants that affect function without altering abundance. For example, in this study, PGD was found to mediate a strong *Akr1e1* distal pQTL in CC and DO mice. However, *Zfp985* has been identified previously as the mediator based on gene expression in CC mice[47] but was not detected at the protein level in this study.

We identified candidate mediators for each of the 20 shared distal pQTLs (Figure 2I), of which only four had best candidate mediators that differed between CC and DO mice. For example,

TUBGCP3 is the strongest candidate mediator of the distal pQTL for *Tubg1* in CC mice, but NAXD is the strongest candidate mediator in DO mice (Figures 2L and 2M). Given that *Tubg1* and *Tubgcp3* (as well as *Tubgcp2*, which also has a co-mapping distal pQTL) are members of the tubulin superfamily, TUBGCP3 is a strong functional candidate, suggesting that NAXD is likely a false positive mediator in DO mice.

We also examined candidate mediators for all distal pQTLs that were detected (FDR < 0.1) in only one of the populations and evaluated the corresponding pQTL status (stringent, lenient, or not detected) and mediation status (e.g., same or different) in the other population. We found the same candidate mediator between CC and DO mice for 21 of 460 distal pQTLs mapped across both populations, suggesting that mediation is more accurate for strong distal pQTLs that are detected in both populations (Figure S1L).

## Drivers of variation in the co-abundance of protein complexes

Members of protein complexes exhibit varying degrees of co-abundance, to which we refer as cohesiveness of the complex. We quantify cohesiveness as the median Pearson correlation between complex members.[18] A high level of cohesiveness suggests that co-regulation of protein abundances across a complex is maintaining stoichiometry. We found that individual proteins that are part of a complex[48–50] (Table S6) are less heritable and have fewer pQTLs than proteins that are not part of a complex (Figures S4A–S4D). We evaluated the extent to which members of protein complexes were inter-correlated as well as how genetic factors and sex contribute to variation in their joint abundance. To assess the contributions from genetic factors and sex, we performed principal-component analysis (PCA) on the abundances of proteins for each protein complex[51] and took the first principal component (PC1) as a summary (STAR Methods). We then estimated heritability and the proportion of variation explained by sex for each complex PC1.

Protein complex cohesiveness was correlated between CC and DO mice ($r = 0.68$, $p < 2.2e-16$) (Figures 3A and 3B), and within each population, it is correlated with complex heritability ($r = 0.28$, $p = 1.2e-4$ in CC mice and $r = 0.12$, $p = 0.11$ in DO mice) (Figure S4E), suggesting that cohesiveness reflects some degree of shared genetic regulation in CC mice. Complex heritability is consistently higher in CC mice than in DO mice (115 of 155 complexes, 74.2%) and is uncorrelated with complex heritability in DO mice ($r = 0.09$, $p = 0.26$) (Figures 3C and 3D). This lack of correlation between heritability of complexes for the CC and DO mice contrasts with the highly correlated heritability of individual proteins ($r = 0.44$, $p < 2.2e-16$). The proportion of variation in complex abundance (as summarized by PC1) explained by sex is correlated between CC and DO mice ($r = 0.70$, $p < 2.2e-16$; Figures 3E and 3F). Protein complexes that have been shown previously to have sex-specific abundance in DO mice,[18] such as eIF2B, were confirmed in CC mice.

## Genetic and stoichiometric regulation of the exosome

The exosome complex provides a striking example of complex-level genetic regulation (Figures 4A–4E). It had the highest complex heritability in CC mice (87.8% [81.7%–90.8%]) but very low

**Cell Genomics**
**Article**

**CellPress**

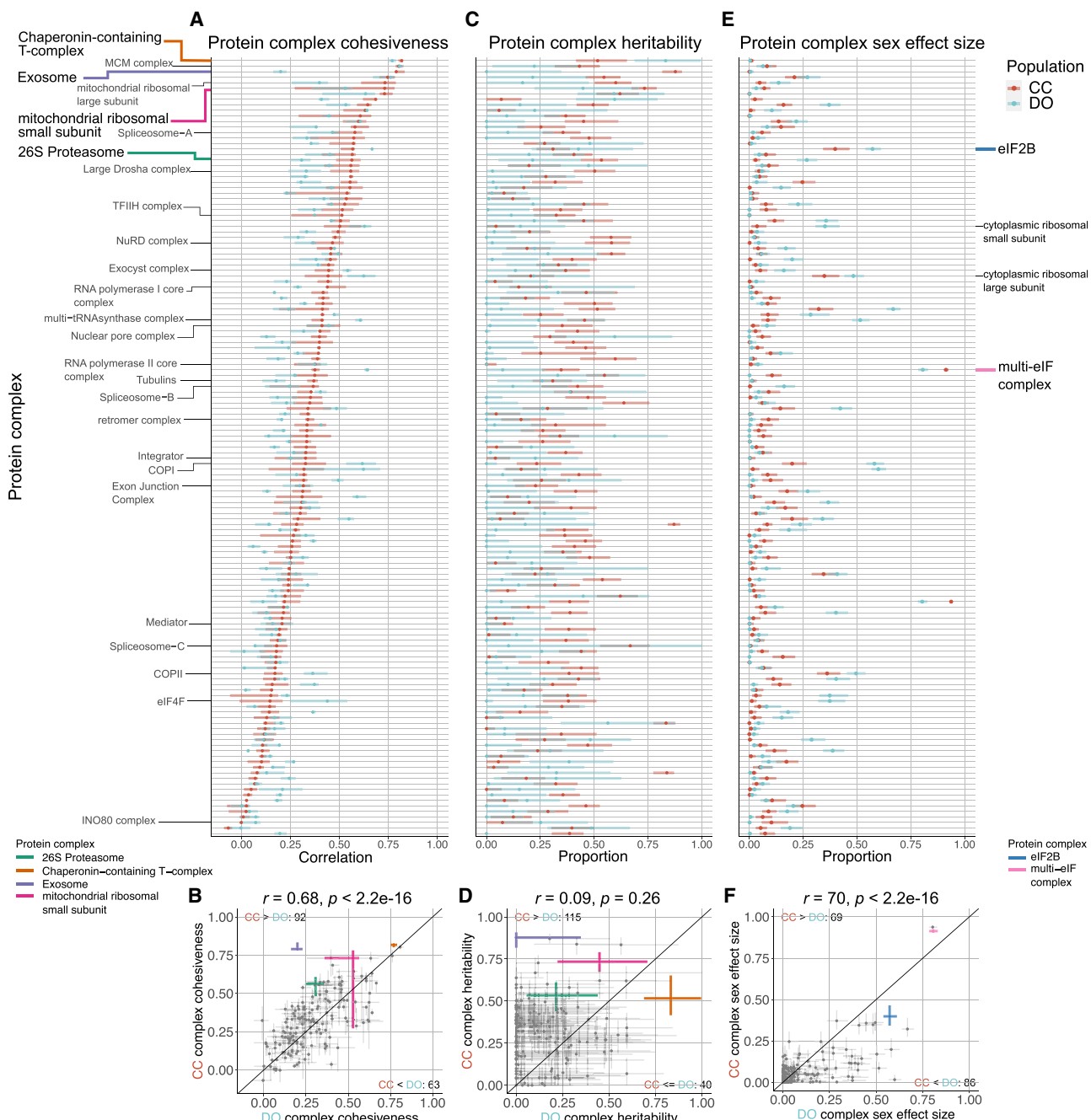

**Figure 3. Genetic and sex effects on protein complexes**

(A and B) Complex cohesiveness, the median pairwise Pearson correlation among members, for CC (red) and DO (blue) mice for 163 protein complexes. Intervals represent the interquartile range, and points represent the overall median.

(C–F) Complex heritability (C and D) and complex sex effect size (E and F), the proportion of variance in PC1 explained by sex, are estimated using the first principal component (PC1) from each of the protein complexes. Intervals represent 95% subsample intervals (STAR Methods). The exosome, CCT complex, 26S proteasome, and MRSS are highlighted as examples of protein complexes with unique genetic effects patterns (Figures 4, 5, 6, S5, and S6). The multi-eIF complex and eIF2B are highlighted as complexes with large sex differences in CC and DO mice. The identity line is included for reference. Pearson correlation coefficients (*r*) between CC and DO mice and corresponding p values included.

See also Figure S4 and Table S6.

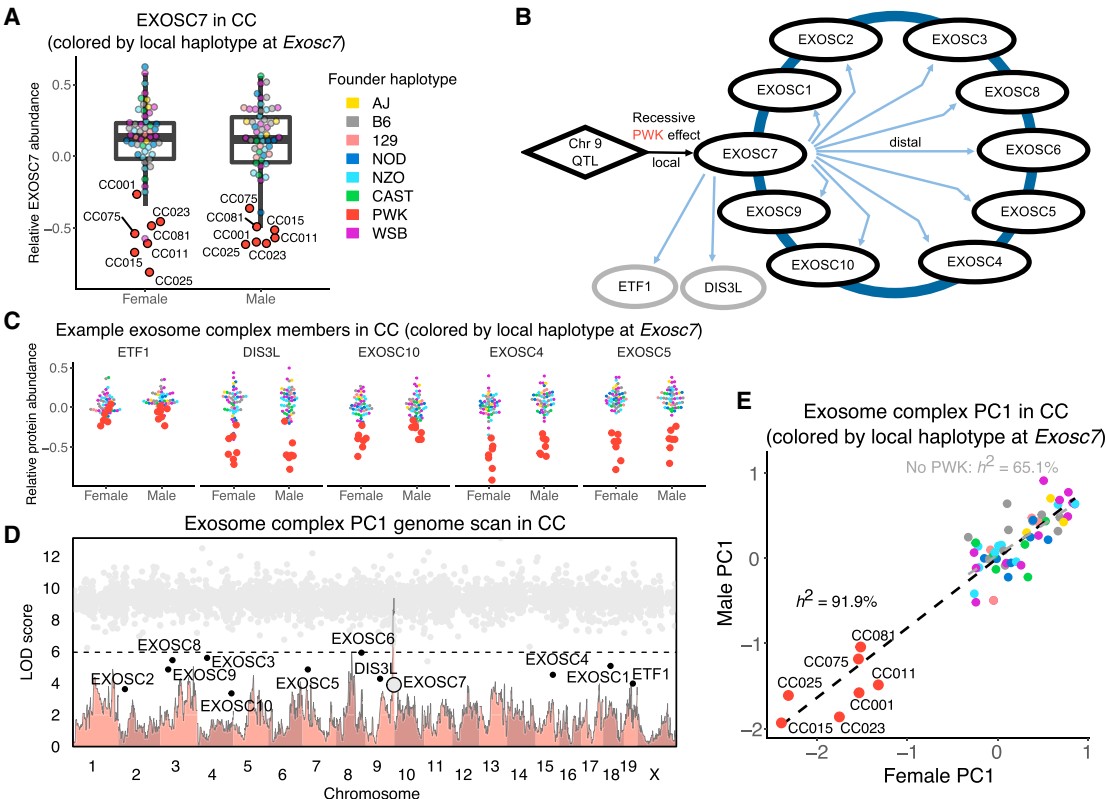

**Figure 4. Stoichiometry-driven genetic effects on the exosome**

(A) Abundance of EXOSC7 in male and female CC mice. Points are color coded to indicate their haplotypes at the *Exosc7* locus. The seven CC strains with low abundance of EXOSC7 have the PWK haplotype (red points) at the *Exosc7* locus and are highlighted.

(B) The effects of a chromosome 9 QTL are mediated through EXOSC7 to affect variation in proteins of the exosome and two functionally related proteins, DIS3L and ETF1.

(C) Abundances of proteins with distal pQTLs at the *Exocs7* locus for female and male CC mice. Points are colored by the founder haplotype at Exosc7.

(D) The genome scan for PC1 of protein abundances in the exosome complex for CC mice (light red) overlayed with the mediation conditional LOD scores (gray dots). Mediation scores were evaluated for all proteins genome-wide. Proteins with low mediation scores are labeled. The horizontal line at a LOD score of 6 is included as reference point across genome scans.

(E) The exosome PC1 plotted as males versus females for the CC strains. Points are colored by the founder haplotype at *Exosc7*. The black dashed line is the best fit line between males and females for the complex PC1, based on all 58 CC strains. The gray dashed line shows the best fit line excluding the seven CC strains with the PWK haplotype at *Exosc7*.

See also Figure S5.

heritability in DO mice (0.0% [0.0%–34.8%]). Low abundance of EXOSC7 in the presence of a local PWK genotype in CC mice appears to be the main driver of exosome complex abundance. Seven CC strains are homozygous for the PWK haplotype at this locus, whereas in the DO cohort, there were no mice homozygous for the PWK haplotype (Figure S5D). Among the DO mice with one copy of the PWK haplotype, there is no reduction in the abundance of the exosome complex, which suggests that the PWK haplotype effect on *Exosc7* is recessive (Figure S5E). We also note that inbred founder PWK mice have low EXOSC7 abundance (Figure S5F). Mediation analysis identifies EXOSC7 as a candidate distal regulator of the complex as well as the functionally related genes *Dis3l* and *Etf1*. These two genes were not included in the complex annotations, but our findings suggest that they are maintained in stoichiometric balance with the annotated complex members. The complex heritability (with *Dis3l* and *Etf1* now included) was 91.9%, and after removing the seven CC

strains with the PWK haplotype at *Exosc7*, it was reduced but still high at 65.1% (Figure 4E), indicating the presence of additional genetic factors that affect the abundance of the exosome.

## Secondary genetic effects on the chaperonin-containing T (CCT) complex

Previously we reported that the CCT complex was stoichiometrically regulated by low abundance of CCT6A when the NOD haplotype is present.[1] The CCT complex (Figures S6) has high heritability in DO mice (83.2% [68.8%–99.6%]) and in CC mice (51.6% [41.6%–65.2%]) (Figures 3C and 3D). The DO sample includes 19 (9.9%) mice homozygous for the NOD haplotype (Figure S6C). The CC strains, six of which are homozygous for NOD at the *Cct6a* locus (Figure S6D), replicate this distal pQTL for the complex members *Cct4*, *Cct5*, *Cct8*, and *Tcp1*, although CCT6a itself was not detected in the CC samples. The effect of the pQTL at *Cct6a* in CC mice drives less of the overall variation in the CCT

complex. A secondary genetic effect mediated through CCT4 is revealed in CC mice corresponding to high abundances in the presence of NZO or PWK haplotypes at *Cct4*. The complex heritability (including all complex members) was 56.8%. After excluding CC strains with the NOD haplotype at *Cct6a* and NZO or PWK haplotypes at *Cct4*, heritability of the complex abundance is 44.9% (Figure S6H), indicating that, as with the exosome, additional genetic effects contribute to CCT complex abundance.

### Independent genetic effects on the subcomplexes of the 26S proteasome

The 26S proteasome is composed of a 20S proteasome catalytic core (PSMA and PSMB proteins) that, in the constitutive form, incorporates subunits PSMB5, PSMB6, and PSMB7 and is capped by two 19S regulators (composed of the PSMC and PSMD proteins). The constitutive form can be modified by replacing the PSMB subunits with the three immunoproteasome-inducible subunits (PSMB8, PSMB9, and PSMB10) and the 19S regulators with the 11S regulators, composed of PSME proteins[52] (Figure 5A). The immunoproteasome is a highly efficient form of the proteasome that is predominantly, but not exclusively, expressed in immune cells.[53]

This alternation between two different forms of the proteasome is apparent in the correlations among the inducible and immune components in the CC, DO, and founder strains (Figures 5B, 5C, and 5I). Individual mice appear to predominantly express one of the proteasome forms, as suggested by the anti-correlation between the constitutive and inducible components. Across the founder strains, this dynamic appears to be regulated genetically, with the WSB, AJ, B6, and NZO strains expressing more immunoproteasome components and the other founder strains expressing more of the constitutive components (Figures S7). In CC and DO mice, we identified genetic variation that controls the balance between PSMB6 (constitutive) and PSMB9 (immunoproteasome) (Figures 5F and 5G). The WSB haplotype at *Psmb9* drives higher PSMB9 abundance as well as lower abundance of its constitutive analog PSMB6, confirmed through mediation analysis (Figures 5H and 5I). The *Psmb9* local pQTL only appears to drive the balance between PSMB9 and PSMB6 and does not directly affect the other interchangeable members of the proteasome (PSMB5, PSMB7, PSMB8, and PSMB10), which do not map strong pQTLs. The distinct correlation patterns among the interchangeable components suggests that they are still co-regulated across the three populations.

Some of the other non-interchangeable components of the 26S proteasome are regulated by genetic variation independent of the *Psmb9* locus. We identified a strong local pQTL for *Psmd9* that is present in CC and DO mice and does not affect other members of the 19S regulator (Figures 5D and 5E), which explains the lack of cohesiveness of PSMD9 with the rest of the proteasome, as noted previously in these DO mice.[18]

### Polygenic regulation of the mitochondrial ribosomal small subunit (MRSS)

The MRSS is highly cohesive in CC and DO mice (Figures 3A, 3B, 6A, and 6B). Complex heritability is also high in CC (73.4% [67.3%–79.1%]) and DO (44.8% [22.2%–70.6%]) mice. Despite its high complex heritability, we detect few pQTLs for individual

members of the complex. One exception is *Auh*, which has a local pQTL in CC and DO mice (Figures 6E and 6F). AUH is not a core member of the MRSS but has been associated with it[54] and is included in the annotations. AUH's local pQTL and lack of cohesion with the core MRSS proteins indicate that it is regulated separately from the core MRSS. Similarly, RPS15 and PPME1 were annotated with the complex but not correlated with the core proteins, suggesting that they are also not co-regulated with the MRSS. MRPS27 and MRPS28, on the other hand, were missing from the annotations and are thus not included in Figure 6, but we found them to be highly cohesive with the core MRSS. For the core MRSS proteins, CC strains have an overall abundance that is highly strain specific, as represented in the complex PC1 and even for individual proteins ($r$ = 0.82 between males and females for the complex PC1 and $r$ = 0.76 for MRPS7). Furthermore, the variation across CC strains is highly continuous (Figures 6C and 6D). This distribution contrasts with the bimodal abundance pattern for the exosome complex, which is driven mostly by a single strong pQTL (Figure 4E), suggesting that many loci with small effects influence the overall abundance of the MRSS.

### CC strain-specific variation affects protein abundance

Inbred mouse strains accumulate mutations that can have phenotypic consequences.[55–58] New mutations have arisen and become fixed in the CC strains[29,30] (Figure 1A), which may affect protein abundance. We confirmed functional effects on protein abundance for genes with CC strain-specific deletions, including the 80-kbp deletion in CC026 that includes the *C3* coding gene and a 15-bp deletion in the *Itgal* gene (Figures 7A and 7B) that occurred in CC042 and has been shown to increase susceptibility to tuberculosis[59] and *Salmonella*.[60]

We estimated strain-specific abundance levels for every protein detected in CC mice and identified CC strains where the male and female had a distinctly low or high abundance of a given protein (|$Z$ score | > 2.5; STAR Methods), which we refer to as strain-specific protein outliers. In total, we identified 5,907 strain-specific protein outliers representing 4,267 proteins across all 58 CC strains. Of these, 67 strain-specific protein outliers occur in strains with a unique genetic variant in or near the coding gene.[30] Furthermore, not all genes with a strain-specific protein outlier and matching strain-specific genetic variants were associated with low protein abundance, as is the case for the deletions; we observed high abundance associated with strain-specific variants, such as *Sash1*, which harbors a novel SNP allele in CC058 (Figure 7C).

Outlying protein abundance patterns specific to a CC strain may represent larger biological pathway dynamics that result from the strain's genetic background. We defined sets of proteins for each strain based on having low or high strain-specific protein outliers and observed strain-specific enrichments for biological functions based on GO and KEGG pathway terms (Table S7). In CC013, we observed increased abundance in proteins related to the innate immune system (Figure 7D), leukocytes, and other immune system-related GO terms. CC013 possesses a unique SNP allele in *Hcls1* that was associated with increased HCLS1 abundance (Figure 7E), a gene involved in myeloid leukocyte differentiation that may contribute to the high abundance of these immune-related proteins. During the process of tissue collection, it was noted that

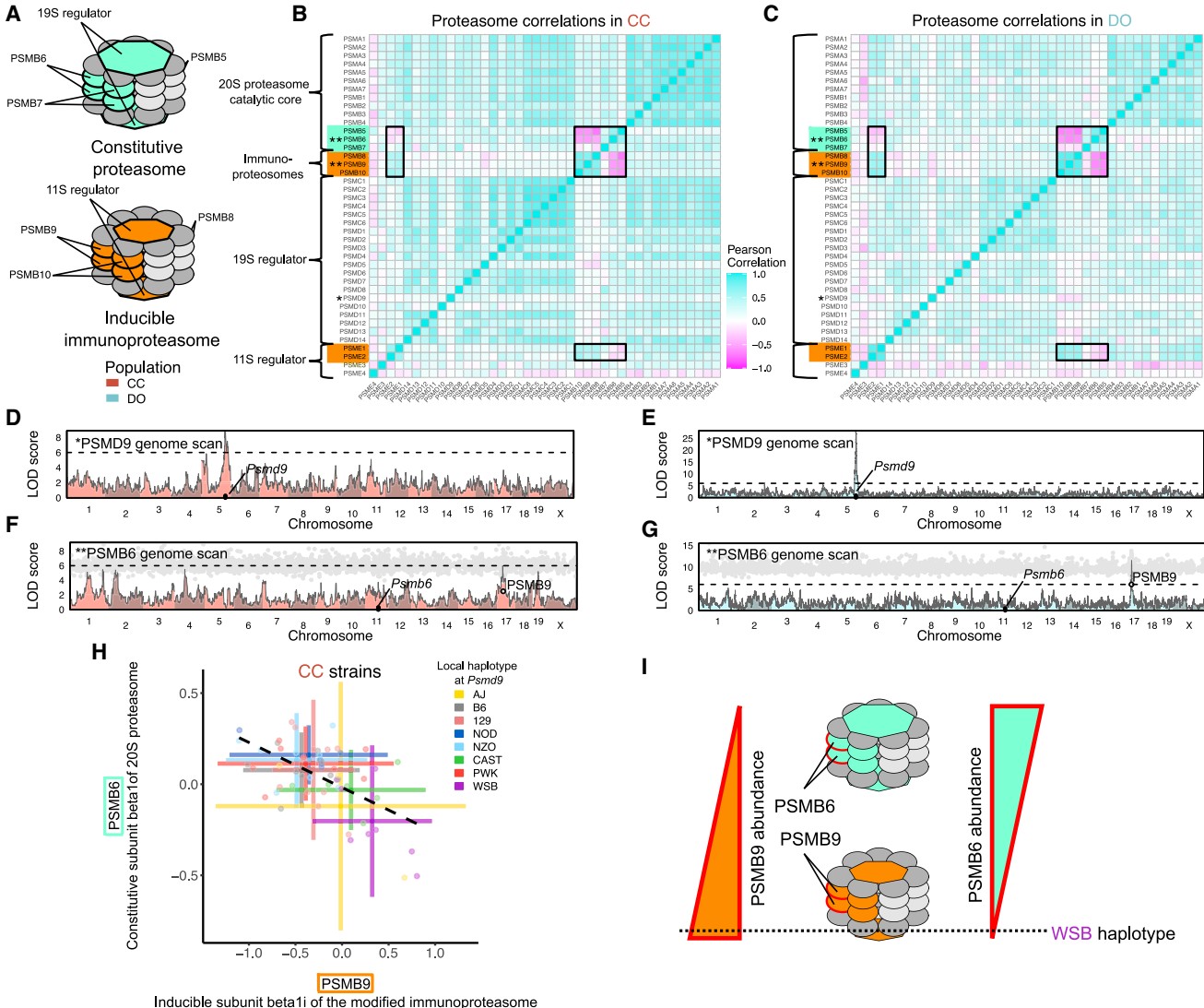

**Figure 5. Genetic control of the constitutive and inducible forms of the 26S proteasome**
(A) The 26S proteasome is composed of multiple subcomplexes: the 20S proteasome catalytic core (PSMA and PSMB proteins) and 19S regulator (PSMC and PSMD proteins) for the constitutive form and the inducible immunoproteasomes (PSMB8, PSMB9, and PSMB10) with their 11S regulator (PSME proteins).
(B and C) The Pearson correlations of the 26S proteosome proteins in (B) CC and (C) DO mice. Black boxes were added to highlight correlations between the constitutive and inducible components.
(D and E) PSMD9 genome scan in (D) CC and (E) DO mice. The horizontal line at a LOD score of 6 is included as reference point across genome scans.
(F and G) PSMB6 genome scan in (F) CC and (G) DO mice overlaid with mediation conditional LOD scores (gray dots) for all proteins, with PSMB9 highlighted.
(H) The abundance of PSMB6 is plotted against the abundance of PSMB9 for CC mice. Horizontal and vertical bars represent means ± 2 standard deviations. Points and bars are colored by the founder haplotype at *Psmd9*. The dashed line is the best fit line between PSMB6 and PSMB9.
(I) CC strains with the WSB haplotype at *Psmd9* have greater abundance of PSMB9 relative to its constitutive analog PSMB6.
See also Figure S7.

CC013 had a unique liver phenotype characterized by white granules throughout the tissue. We examined additional mice to confirm this (Figure 7F) and hypothesize that the liver granules are related to an excess of immune-related proteins. Additional CC strains (Figures S8C–S8E) with multiple outlier proteins that are functionally related include CC007 (Figure 7G), which has low- and high-abundance proteins in mitochondrial respiratory complex I. The replenishable inbred CC strains capture these dynamics and allow

deeper interrogation of unique strain-specific networks of functionally related proteins with perturbed abundance and a better understanding of their phenotypic consequences.

## DISCUSSION

We carried out proteomics profiling of mice from the CC and DO strains and their founder strains. Despite the challenges imposed

**Cell Genomics**
Article

**Figure 6. Polygenic regulation of the mitochondrial ribosomal small subunit (MRSS)**

(A and B) The Pearson correlations of the MRSS in (A) CC and (B) DO mice.

(C and D) MRPS7 (*) abundance (C) and PC1 of protein abundances from the MRSS core (black box) (D) plotted as males versus females for the CC strains. The dashed lines represent the best fit lines between males and females for MRPS7 and complex PC1.

(E and F) Genome scans for AUH (**), a protein affiliated with the mitochondrial ribosome that is largely uncorrelated with core members of the complex, reveal a local pQTL detected in (E) CC and (F) DO mice. Horizontal lines at a LOD score of 6 are included as a reference point across genome scans.

by separate experiments and the relative nature of MS proteomics,[61] the data provided consistent results that supported comparative analysis. Genetic regulation of the proteome in the liver is highly conserved across these distinct but related genetic reference populations. The concordance is exceptionally strong for local genetic regulation and sex differences but also for distal genetic effects strongly detected in both populations. Discordance between CC and DO mice can often be attributed to chance differences in allele frequencies or to dominance effects that manifest differently in the inbred versus outbred populations, similar to what has been observed in *Drosophila* for gene expression.[62] Mediation analysis of distal pQTLs identified many of the same candidate mediators in CC and DO mice. Proteins that form complexes are generally less affected by local genetic regulation compared with other proteins. Complexes displayed a wide range of cohesiveness that was more highly conserved across populations than complex heritability; nevertheless, genetic effects on protein complexes can manifest in remarkably different ways. The CC strains enable discovery of extreme strain-specific abundance of individual proteins and of functionally related groups of proteins, which can be recaptured and studied further because inbred strains are replenishable.

## Implications

Protein complex members are often synthesized in proportions that are consistent with stoichiometric balance.[17] Genetic perturbations of one or more member proteins can introduce imbalances that need to be compensated, typically

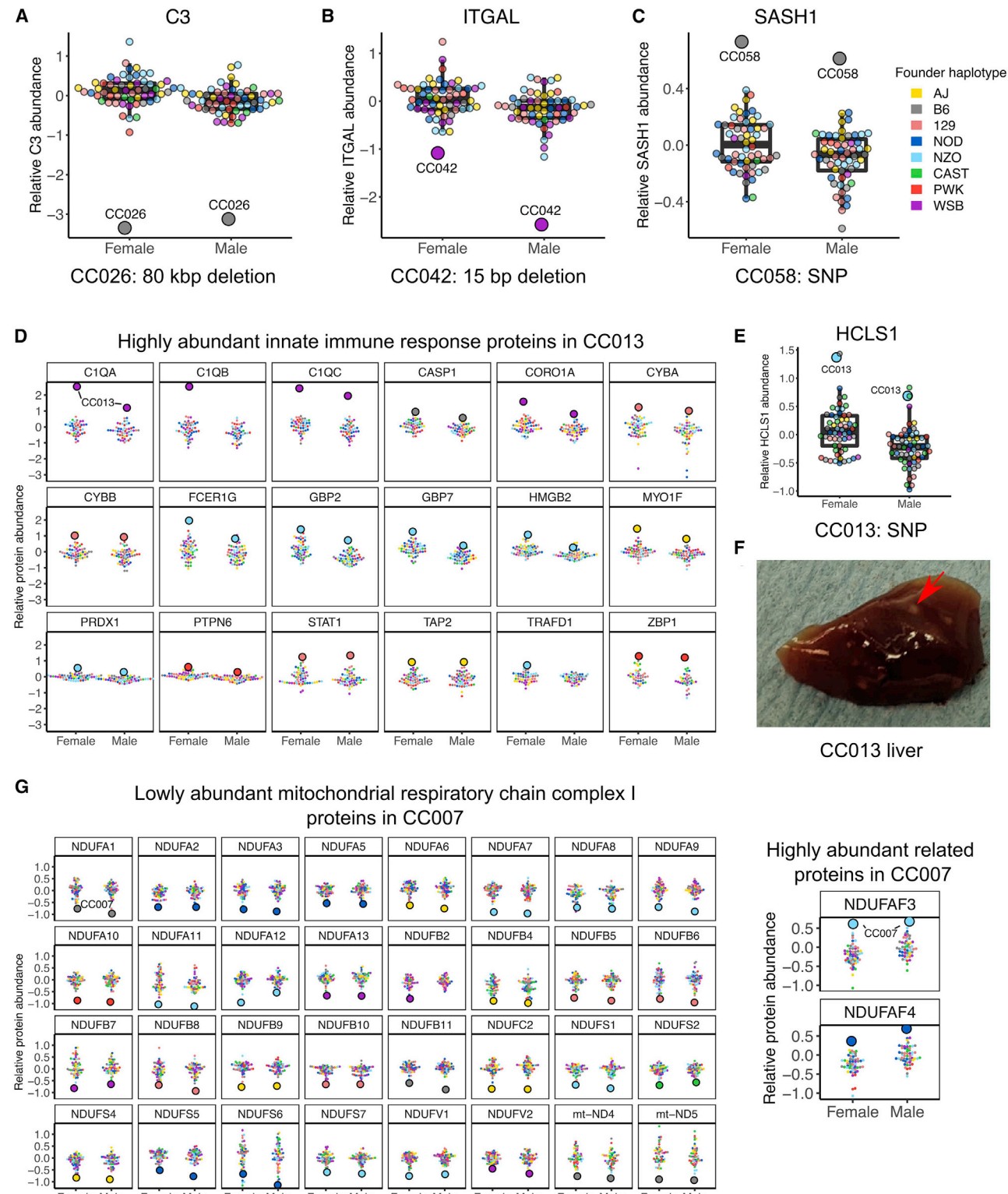

**Figure 7. Strain-specific genetic variants affect protein abundance and influence larger protein networks**

(A–C) Abundance for (A) C3, (B) ITGAL, and (C) SASH1 in female and male CC mice with the outlier strain highlighted.

(D) Abundance of proteins related to innate immune response are shown for female and male CC mice with the outlier strain CC013 indicated.

*(legend continued on next page)*

through degradation pathways that recycle unassembled units of protein complexes. In this way, a genetic variant that severely reduces transcription of a gene and, consequently, the protein abundance of a single complex member can have effects that propagate through the entire complex and be detected as a shared distal pQTL, as was the case for the exosome and CCT complexes. At the other extreme, the abundance of a highly cohesive complex such as the MRSS can be heritable with few or no detectable pQTLs, which is consistent with polygenic regulation by multiple small-effect loci. Even the exosome and CCT complex display significant residual heritability after accounting for their large-effect pQTL, indicating that polygenic effects on complex abundance are pervasive.

We observed some inconsistent results between CC and DO mice using the PC1 to estimate complex heritability and sex effects. This is likely due to differences in how the main axes of variation differ, suggesting that a single summary measure is insufficient to capture the behavior of many complexes. Examination of the entire correlation matrix of protein complexes can reveal a more detailed picture of the regulatory structure, such as internal heterogeneity in the relative balance of components. For example, our analysis of the 26S proteosome reveal known subcomplexes and individual proteins that are regulated independently. In addition, our analysis of the MRSS highlights some shortcomings of current protein complex definitions that can potentially be corrected based on the correlation structure in shotgun proteomics data.

*De novo* mutations specific to individual CC strains are clearly responsible for outlying abundance patterns for some proteins, but we identified a large number of such outliers, and it seems implausible that the majority of these would be due to mutations. Based on the functional similarity of protein outliers within specific strains, we propose that these represent perturbations of interacting networks of proteins, whether they are due to *de novo* mutations or to multi-locus allelic combinations that are fixed in specific CC strains. Epistasis, particularly among interacting proteins, could contribute to these CC strain-specific networks. Regardless of their underlying origin, CC strains with single or functional groups of protein outliers can serve as models for further investigation of biological mechanisms and disease.

### Limitations of study

Our study has insufficient power to reliably detect and characterize small distal genetic effects, which likely contributes to the reduced concordance of distal pQTLs between CC and DO mice. This study also highlights some of the caveats of mediation analysis. Successful mediation analysis requires that the true mediator is present in the data and that its effects are mediated through variation in abundance and not through other functional changes in the protein. A protein that is correlated with the true

but unobserved mediator may be identified incorrectly as a candidate mediator. In addition to unobserved proteins, other factors, such as non-coding RNAs that could mediate distal pQTLs, may not be measured for a given experiment. Proteins with strong local pQTLs can appear to be mediators, as was likely the case for mediation of the *Tubg1* distal pQTL through NAXD in DO mice. Comparison of mediation analysis across independent genetic experiments can correct and refine candidate mediators.

### Outlook

Unbiased profiling of the proteome provides a unique window into the molecular processes that are active in cells and tissues, a view that is complementary to and often more directly relevant to function than transcriptome profiling. Although many proteins are responsive to transcriptional regulation, they can also be regulated by a variety of post-translational mechanisms. Analysis of proteomics data in genetically diverse populations provides causal perturbations in the form of genetic variation that introduce variability in protein abundance across all levels of regulation. Genetic mapping and correlation analyses can identify co-regulated proteins and key drivers that regulate other proteins and protein complexes. Technologies that measure protein abundance are developing rapidly but are already capable of delivering accurate and reproducible data. Together with the demonstrated consistency of genetic effects on proteins across distinct but related mouse resource populations, this suggests that we can extrapolate findings across these genetic reference populations with some confidence. Resource data such as described here can be co-analyzed with future data through meta-analyses; for example, comparing genetic effects across different tissues. These findings suggest that imputation of locally regulated proteins could be an option when direct profiling of proteins is not available.

### STAR★METHODS

Detailed methods are provided in the online version of this paper and include the following:

- KEY RESOURCES TABLE
- RESOURCE AVAILABILITY
  - Lead contact
  - Materials availability
  - Data and code availability
- EXPERIMENTAL MODEL AND SUBJECT DETAILS
  - Mice
- METHOD DETAILS
  - Mouse genotyping, founder haplotype reconstruction, and gene annotations
  - Sample preparation for proteomics analysis
  - Offline basic pH reversed-phase (BPRP) fractionation

---

(E) Abundance for HCLS1 in female and male CC mice with the outlier strain CC013 indicated. Point color corresponds to the founder haplotype at the gene locus of the specified protein.

(F) CC013 has a unique liver phenotype characterized by white granules, indicated with a red arrow.

(G) Abundance of proteins related to the mitochondrial respiratory chain complex l are shown for female and male CC mice with the outlier strain CC007 indicated. See also Figure S8 and Table S7.

- ○ Liquid chromatography and tandem mass spectrometry
- ○ Mass spectra data analysis
- ● QUANTIFICATION AND STATISTICAL ANALYSIS
  - ○ Filtration of peptides that contain polymorphisms
  - ○ Protein abundance estimation from peptides
  - ○ Heritability estimation
  - ○ QTL analysis
  - ○ QTL significance thresholds
  - ○ Defining local/distal status of QTL
  - ○ Consistency of QTL between the CC and DO
  - ○ Consistency of local QTL in the CC with the founder strains
  - ○ Mediation analysis
  - ○ Sex effects on protein abundance analysis
  - ○ Protein complex analysis
  - ○ Strain-specific outlier proteins
- ● ADDITIONAL RESOURCES

## SUPPLEMENTAL INFORMATION

## ACKNOWLEDGMENTS

We thank Kwangbom Choi, Andrew Deighan, and Isabela Gerdes Gyuricza of the Churchill lab for helpful discussions and encouragement throughout the course of this project. We thank Lauren J. Donoghue of the University of North Carolina at Chapel Hill, John W. Keele of the United States Department of Agriculture, Paul L. Maurizio of the University of Chicago, and Bryan C. Quach of the Research Triangle Institute for reading and providing feedback on the manuscript. This work was supported by grant funding from the National Institutes of Health (NIH): F32GM134599 to G.R.K.; U19AI100625, P01AI132130, and R01ES029925 to F.P.-M.d.V. and M.T.F.; R01GM067945 to S.P.G.; and R01GM070683 to G.A.C.

## AUTHOR CONTRIBUTIONS

Conceptualization, M.T.F., S.P.G., and G.A.C.; methodology, G.R.K., T.Z., S.P.G., and G.A.C.; software, G.R.K., D.T.P., and M.V.; investigation, G.R.K. and T.Z.; resources, T.Z., J.A.P., T.A.B., P.H., G.D.S., F.P.-M.d.V., M.T.F., and S.P.G.; data curation, G.R.K., T.Z., and M.V.; writing – original draft, G.R.K., T.Z., and G.A.C.; writing – review & editing and visualization, G.R.K.; supervision, S.C.M., M.T.F., S.P.G., and G.A.C.; funding acquisition, F.P.-M.d.V., M.T.F., S.P.G., and G.A.C.

## DECLARATION OF INTERESTS

The authors declare no competing interests.

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

## STAR★METHODS

### KEY RESOURCES TABLE

| REAGENT or RESOURCE | SOURCE | IDENTIFIER |
|---|---|---|
| **Chemicals, peptides, and recombinant proteins** | | |
| Pierce Protease Inhibitor Tablets | Thermo Fisher | A32963 |
| Pierce Phosphatase Inhibitor Mini Tablets | Thermo Fisher | A32957 |
| Trypsin Protease MS grade, Frozen | Thermo Fisher | 90305R200 |
| Lys-C, Mass Spectrometry Grade | Wako Chemicals | Barcode#4987481427648 |
| TMT10plex Isobaric Label reagent Set plus TMT11-131C Label Reagent | Thermo Fisher | A34808 |
| **Critical commercial assays** | | |
| Pierce BCA Protein Assay Kit | Thermo Fisher | 23227 |
| **Deposited data** | | |
| CC liver proteomics | ProteomeXchange (http://www.proteomexchange.org) | PXD018886 |
| DO and founder strain liver proteomics | Chick et al.[1]; ProteomeXchange (http://www.proteomexchange.org) | PXD002801 |
| Processed data (e.g., proteins, peptides, genotypes) and code to generate all results and figures | https://doi.org/10.6084/m9.figshare.12818717 | N/A |
| CC liver proteomics QTL Viewer | https://qtlviewer.jax.org/viewer/FerrisCC | N/A |
| DO liver proteomics QTL Viewer | https://qtlviewer.jax.org/viewer/SvensonHFD | N/A |
| **Experimental models: Organisms/strains** | | |
| Mouse: A/J | The Jackson Laboratory | JAX: 000646 |
| Mouse: C57BL/6J | The Jackson Laboratory | JAX: 000664 |
| Mouse: 129S1/SvImJ | The Jackson Laboratory | JAX: 002448 |
| Mouse: NOD/ShiLtJ | The Jackson Laboratory | JAX: 001976 |
| Mouse: NZO/HILtJ | The Jackson Laboratory | JAX: 002105 |
| Mouse: CAST/EiJ | The Jackson Laboratory | JAX: 000928 |
| Mouse: PWK/PhJ | The Jackson Laboratory | JAX: 003715 |
| Mouse: WSB/EiJ | The Jackson Laboratory | JAX: 001145 |
| Mouse: J:DO | The Jackson Laboratory | JAX: 009376 |
| Mouse: CC001/Unc | UNC Systems Genetics Core | N/A |
| Mouse: CC002/Unc | UNC Systems Genetics Core | N/A |
| Mouse: CC003/Unc | UNC Systems Genetics Core | N/A |
| Mouse: CC004/TauUnc | UNC Systems Genetics Core | N/A |
| Mouse: CC005/TauUnc | UNC Systems Genetics Core | N/A |
| Mouse: CC006/TauUnc | UNC Systems Genetics Core | N/A |
| Mouse: CC007/Unc | UNC Systems Genetics Core | N/A |
| Mouse: CC008/GeniUnc | UNC Systems Genetics Core | N/A |
| Mouse: CC009/UncJ | UNC Systems Genetics Core | N/A |
| Mouse: CC010/GeniUnc | UNC Systems Genetics Core | N/A |
| Mouse: CC011/Unc | UNC Systems Genetics Core | N/A |
| Mouse: CC012/GeniUnc | UNC Systems Genetics Core | N/A |
| Mouse: CC013/GeniUnc | UNC Systems Genetics Core | N/A |
| Mouse: CC015/Unc | UNC Systems Genetics Core | N/A |
| Mouse: CC016/GeniUnc | UNC Systems Genetics Core | N/A |
| Mouse: CC019/TauUnc | UNC Systems Genetics Core | N/A |

*Continued*

| REAGENT or RESOURCE | SOURCE | IDENTIFIER |
|---|---|---|
| Mouse: CC021/Unc | UNC Systems Genetics Core | N/A |
| Mouse: CC023/GeniUnc | UNC Systems Genetics Core | N/A |
| Mouse: CC024/GeniUnc | UNC Systems Genetics Core | N/A |
| Mouse: CC025/GeniUnc | UNC Systems Genetics Core | N/A |
| Mouse: CC026/GeniUnc | UNC Systems Genetics Core | N/A |
| Mouse: CC027/GeniUnc | UNC Systems Genetics Core | N/A |
| Mouse: CC029/Unc | UNC Systems Genetics Core | N/A |
| Mouse: CC030/GeniUnc | UNC Systems Genetics Core | N/A |
| Mouse: CC031/GeniUnc | UNC Systems Genetics Core | N/A |
| Mouse: CC032/GeniUnc | UNC Systems Genetics Core | N/A |
| Mouse: CC033/GeniUnc | UNC Systems Genetics Core | N/A |
| Mouse: CC035/Unc | UNC Systems Genetics Core | N/A |
| Mouse: CC036/Unc | UNC Systems Genetics Core | N/A |
| Mouse: CC037/TauUnc | UNC Systems Genetics Core | N/A |
| Mouse: CC038/GeniUnc | UNC Systems Genetics Core | N/A |
| Mouse: CC039/Unc | UNC Systems Genetics Core | N/A |
| Mouse: CC040/TauUnc | UNC Systems Genetics Core | N/A |
| Mouse: CC041/TauUnc | UNC Systems Genetics Core | N/A |
| Mouse: CC042/GeniUnc | UNC Systems Genetics Core | N/A |
| Mouse: CC043/GeniUnc | UNC Systems Genetics Core | N/A |
| Mouse: CC044/Unc | UNC Systems Genetics Core | N/A |
| Mouse: CC045/GeniUnc | UNC Systems Genetics Core | N/A |
| Mouse: CC046/Unc | UNC Systems Genetics Core | N/A |
| Mouse: CC049/TauUnc | UNC Systems Genetics Core | N/A |
| Mouse: CC051/TauUnc | UNC Systems Genetics Core | N/A |
| Mouse: CC053/Unc | UNC Systems Genetics Core | N/A |
| Mouse: CC055/TauUnc | UNC Systems Genetics Core | N/A |
| Mouse: CC057/Unc | UNC Systems Genetics Core | N/A |
| Mouse: CC058/Unc | UNC Systems Genetics Core | N/A |
| Mouse: CC059/TauUnc | UNC Systems Genetics Core | N/A |
| Mouse: CC060/Unc | UNC Systems Genetics Core | N/A |
| Mouse: CC061/GeniUnc | UNC Systems Genetics Core | N/A |
| Mouse: CC062/Unc | UNC Systems Genetics Core | N/A |
| Mouse: CC071/TauUnc | UNC Systems Genetics Core | N/A |
| Mouse: CC072/TauUnc | UNC Systems Genetics Core | N/A |
| Mouse: CC075/Unc | UNC Systems Genetics Core | N/A |
| Mouse: CC078/TauUnc | UNC Systems Genetics Core | N/A |
| Mouse: CC079/TauUnc | UNC Systems Genetics Core | N/A |
| Mouse: CC080/TauUnc | UNC Systems Genetics Core | N/A |
| Mouse: CC081/Unc | UNC Systems Genetics Core | N/A |
| Mouse: CC082/Unc | UNC Systems Genetics Core | N/A |
| Software and algorithms | | |
| Bioconductor | Bioconductor | https://bioconductor.org; RRID: SCR_006442 |
| clusterProfiler | Yu et al.[63] | https://bioconductor.org/packages/release/bioc/html/clusterProfiler.html; RRID: SCR_016884 |
| ensimplR | https://github.com/churchill-lab/ensimplR | N/A |

*(Continued on next page)*

**Cell Genomics**
Article

*Continued*

| REAGENT or RESOURCE | SOURCE | IDENTIFIER |
|---|---|---|
| evd | https://cran.r-project.org/web/packages/evd | N/A |
| intermediate | https://github.com/churchill-lab/intermediate | N/A |
| intermediate2 | https://github.com/duytpm16/intermediate2 | N/A |
| lme4 | Bates et al.[64] | https://cran.r-project.org/web/packages/lme4/index.html; RRID: SCR_015654 |
| pcaMethods | Stacklies et al.[51] | https://www.bioconductor.org/packages/release/bioc/html/pcaMethods.html |
| QTL Viewer webtool | https://github.com/churchill-lab/qtlapi | N/A |
| R | The R Project | https://www.r-project.org; RRID: SCR_001905 |
| R/qtl2 | Broman et al.[65] | https://github.com/rqtl/qtl2; RRID: SCR_018181 |
| Other | | |
| Complex Database | Ori et al.[49] | http://doi.org/10.1186/s13059-016-0912-5; Table S6 |
| DO founder haplotype dosages (genoprobs) | Chick et al.[1] | http://doi.org/10.1038/nature18270 |
| SQLite CC founder variant database | https://doi.org/10.6084/m9.figshare.5280229.v3 | N/A |
| Waters 100mg Sep-Pak | Waters | WAT036820 |
| Orbitrap Fusion | Thermo Fisher | N/A |
| Orbitrap Fusion Lumos | Thermo Fisher | N/A |

## RESOURCE AVAILABILITY

### Lead contact
Further information and requests for resources should be directed to and will be fulfilled by the lead contact Gary Churchill (gary.churchill@jax.org).

### Materials availability
The CC strains used in this study (key resources table) are available from the UNC Systems Genetics Core (https://csbio.unc.edu/CCstatus/index.py?run=availableLines). Many of the strains are also available from the Jackson Laboratory.

### Data and code availability
The mass-spec proteomics data for the CC liver samples reported here have been deposited in ProteomeXchange (http://www.proteomexchange.org/) via the PRIDE partner repository (ProteomeXchange: PXD018886). This study also makes use of existing, publicly available liver proteomics data from the DO and founder strains (ProteomeXchange: PXD002801).[1]

All analyses were performed using the R statistical programming language (v3.6.1).[2] The analysis pipeline used to generate the results, starting from the raw data, scripts to process the raw data, the processed data, and scripts to analyze the processed data and generate the figures, has been made publicly available (figshare: https://doi.org/10.6084/m9.figshare.12818717).

All processed data and pQTL results are also available for download and interactive analysis from the QTL Viewer webtool (https://github.com/churchill-lab/qtlapi) for both the CC (https://qtlviewer.jax.org/viewer/FerrisCC) and DO (https://qtlviewer.jax.org/viewer/SvensonHFD).

## EXPERIMENTAL MODEL AND SUBJECT DETAILS

### Mice
We received pairs of young mice from 58 CC strains from the UNC Systems Genetics Core Facility between the summer of 2018 and early 2019. Mice were singly housed upon receipt until eight weeks of age. More information regarding the CC strains can be found at https://csbio.unc.edu/CCstatus/index.py?run=availableLines.

Mouse studies were performed in strict accordance with the recommendations in the Guide for the Care and Use of Laboratory Animals of the National Institutes of Health. All mouse studies at University of North Carolina at Chapel Hill (UNC) (Animal Welfare Assurance #A3410-01) were performed using protocols approved by the UNC Institutional Animal Care and Use Committee (IACUC) in a manner designed to minimize pain and suffering in infected animals. Any animals that exhibited severe disease signs was euthanized immediately in accordance with IACUC approved endpoints. Mice were kept on a 12h:12h light:dark cycle, with temperature maintained between 68°F and 74°F and 30% and 70% humidity. Mice were provided unrestricted access to food (LabDiet(R) Select Verified Rodent Diets 5V0F - Select Rodent 50 IF/6F auto) and water. Mice were fasted for 6 h before euthanasia and tissue collection.

## METHOD DETAILS

### Mouse genotyping, founder haplotype reconstruction, and gene annotations

The 116 CC mice were genotyped on the Mini Mouse Universal Genotyping Array (MiniMUGA), which includes 11,125 markers.[66] Founder haplotypes were reconstructed using a Hidden Markov Model (HMM), implemented in the qtl2 R package,[65] using the "risib8" option for an eight founder recombinant inbred panel. Heterozygous genotypes were omitted, and haplotype reconstructions are limited to homozygous states, smoothing over a small number of residual heterozygous sites that remain in the CC mice. The genotyping and haplotype reconstruction for the DO mice were previously described;[1] briefly, genotyping was performed on MegaMUGA (57,973 markers),[67] and founder haplotypes were reconstructed using the DOQTL R package.[68]

Ensembl version 91 gene and protein annotations were used in the CC, whereas version 75 was previously used in the DO and founder strains data. If the gene symbol or gene ID differed for a protein ID between versions 75 and 91, we updated them to version 91 in the DO and founder strains. When comparing results (e.g., heritability, sex effects, and pQTLs) between the CC, DO, or founder strains, we merged based on protein ID. For comparing the more complicated mediation analysis, we allowed matches based on mediator gene symbol rather than mediator protein ID if the target protein IDs matched.

### Sample preparation for proteomics analysis

We analyzed liver tissue in the CC to match the previously collected liver data in the DO and founder strains. Sample preparation and mass spectr (MS) analysis for the DO and founder strains were previously described.[1] Singly housed CC mice had their food removed six hours prior to euthanasia and tissue harvest. Tissues were dissected, weighed, and snap frozen in liquid nitrogen. Pulverized CC liver tissue were syringe-lysed in 8 M urea and 200 mM EPPS pH 8.5 with protease inhibitor and phosphatase inhibitor. BCA assay was performed to determine protein concentration of each sample. Samples were reduced in 5 mM TCEP, alkylated with 10 mM iodoacetamide, and quenched with 15 mM DTT. 200 μg protein was chloroform-methanol precipitated and re-suspended in 200 μL 200 mM EPPS pH 8.5. The proteins were digested by Lys-C at a 1:100 protease-to-peptide ratio overnight at room temperature with gentle shaking. Trypsin was used for further digestion for 6 hours at 37°C at the same ratio with Lys-C. After digestion, 50 μL of each sample were combined in a separate tube and used as the 11th sample in all 12 tandem mass tag (TMT) 11plex. 100 μL of each sample were aliquoted, and 30 μL acetonitrile (ACN) was added into each sample to 30% final volume. 200 μg TMT reagent (126, 127N, 127C, 128N, 128C, 129N, 129C, 130N, 130C, 130N, and 131C) in 10 μL ACN was added to each sample. After 1 hour of labeling, 2 μL of each sample was combined, desalted, and analyzed using MS. Total intensities were determined in each channel to calculate normalization factors. After quenching using 0.3% hydroxylamine, 11 samples were combined in 1:1 ratio of peptides based on normalization factors. The mixture was desalted by solid-phase extraction and fractionated with basic pH reversed phase (BPRP) high performance liquid chromatography (HPLC), collected onto a 96 well plate and combined for 24 fractions in total. Twelve fractions were desalted and analyzed by liquid chromatography-tandem mass spectrometry (LC-MS/MS).

### Offline basic pH reversed-phase (BPRP) fractionation

We fractionated the pooled TMT-labeled peptide sample using BPRP HPLC.[69] We used an Agilent 1200 pump equipped with a degasser and a photodiode array (PDA) detector. Peptides were subjected to a 50-min linear gradient from 5% to 35% acetonitrile in 10 mM ammonium bicarbonate pH 8 at a flow rate of 0.6 mL/min over an Agilent 300Extend C18 column (3.5 μm particles, 4.6 mm ID, and 220 mm in length). The peptide mixture was fractionated into a total of 96 fractions, which were consolidated into 24, from which 12 non-adjacent samples were analyzed.[70] Samples were subsequently acidified with 1% formic acid and vacuum centrifuged to near dryness. Each consolidated fraction was desalted via StageTip, dried again via vacuum centrifugation, and reconstituted in 5% acetonitrile, 5% formic acid for LC-MS/MS processing.

### Liquid chromatography and tandem mass spectrometry

Mass spectrometric data were collected on an Orbitrap Fusion Lumos mass spectrometer coupled to a Proxeon NanoLC-1200 UHPLC. The 100 μm capillary column was packed with 35 cm of Accucore 50 resin (2.6 μm, 150Å; ThermoFisher Scientific). Peptides were separated using a 2.5 h gradient of 9∼35% acetonitrile gradient in 0.125% formic acid with a flow rate of ∼400nl min−1. The scan sequence began with an MS1 spectrum (Orbitrap analysis, resolution 120,000, 350−1400 Th, automatic gain control (AGC) target 5E5, maximum injection time 50 ms). SPS-MS3 analysis was used to reduce ion interference.[71,72] The top 10 precursors were then selected for MS2/MS3 analysis. MS2 analysis consisted of collision-induced dissociation (CID), quadrupole ion trap analysis, automatic gain control (AGC) 1E4, NCE (normalized collision energy) 35, $q$-value < 0.25, maximum injection time

60 ms), and isolation window at 0.5. Following acquisition of each MS2 spectrum, we collected an MS3 spectrum in which multiple MS2 fragment ions are captured in the MS3 precursor population using isolation waveforms with multiple frequency notches. MS3 precursors were fragmented by HCD and analyzed using the Orbitrap (NCE 65, AGC 3E5, maximum injection time 150 ms, resolution was 50,000 at 400 Th).

### Mass spectra data analysis

Mass spectra were processed using a Sequest-based pipeline.[73] Spectra were converted to mzXML using a modified version of ReAdW.exe. Database search included all entries from an indexed Ensembl database version 90 (downloaded:10/09/2017). This database was concatenated with one composed of all protein sequences in the reversed order. Searches were performed using a 50 ppm precursor ion tolerance for total protein level analysis. The product ion tolerance was set to 0.9 Da. TMT tags on lysine residues, peptide N termini (+229.163 Da), and carbamidomethylation of cysteine residues (+57.021 Da) were set as static modifications, while oxidation of methionine residues (+15.995 Da) was set as a variable modification.

Peptide-spectrum matches (PSMs) were adjusted to FDR < 0.01.[74,75] PSM filtering was performed using a linear discriminant analysis (LDA), as described previously,[73] while considering the following parameters: XCorr, ΔCn, missed cleavages, peptide length, charge state, and precursor mass accuracy. For TMT-based reporter ion quantitation, we extracted the summed signal-to-noise (S:N) ratio for each TMT channel and found the closest matching centroid to the expected mass of the TMT reporter ion. For protein-level comparisons, PSMs were identified, quantified, and collapsed to a peptide FDR < 0.01 and then collapsed further to a final protein-level FDR < 0.01, which resulted in a final peptide level FDR < 0.001. Moreover, protein assembly was guided by principles of parsimony to produce the smallest set of proteins necessary to account for all observed peptides. PSMs with poor quality, MS3 spectra with TMT reporter summed signal-to-noise of less than 100, or having no MS3 spectra were excluded from quantification.[76]

## QUANTIFICATION AND STATISTICAL ANALYSIS

### Filtration of peptides that contain polymorphisms

Peptides that contain polymorphisms are problematic for protein quantification in genetically diverse samples because the variant peptides cannot be quantified simultaneously. Polymorphisms can result in reduced intensity or non-detection events for peptide isoforms that do not match the reference mouse genome. This in turn can affect protein abundance estimation from peptides and can either obscure the signal of a true pQTL or create a false local pQTL. Therefore, we filtered out polymorphic peptides based on the genome sequences of the founder strains. We further confirmed the presence of the expected polymorphisms by examining the distribution of peptide intensities across samples from the founder strains.

To determine whether peptides with polymorphisms matched their expected allele distribution pattern, the peptide data was standardized within batches and adjusted for batch effects. Each peptide was scaled by a sample-specific within-batch scaling factor: $\bar{y}_i^{pep\ k} = \frac{y_i^{pep\ k}}{\theta_i}$, where $y_i^{pep\ k}$ is the intensity of peptide $k$ for mouse $i$, $\theta_i = \frac{\sum_K y_i^{pep\ k}}{\max\limits_{l\,\in\,B[i]}(\sum_K y_l^{pep\ k})}$, $K$ is the set of all peptides measured for mouse $i$, and $B[i]$ is the set of samples included in batch $i$. For the CC samples, a pooled bridge sample was included in each batch and provided an additional standardization across batches: $\tilde{y}_i^{pep\ k} = \log_2\left(\frac{\bar{y}_i^{pep\ k} + 1}{\bar{y}_{b[i]}^{pep\ k} + 1}\right)$, where $b[i]$ represents the bridge sample from the batch of mouse $i$. For the DO and founder strain samples that did not include bridge samples, $\tilde{y}_i^{pep\ k} = \log_2(\bar{y}_i^{pep\ k} + 1)$. A log transformation was applied to peptide intensities.

Batch effects were removed from the processed peptide data using a linear mixed effect model (LMM) fit with the lme4 R package.[64] Peptides unobserved for all samples within a batch were recorded as missing (NA). If greater than 80% of samples were missing for a polymorphic peptide, it was removed from the batch correction step and the subsequent evaluation. The following model was fit to peptide intensity data for the CC mice:

$$\tilde{y}_i^{pep\ k} = \mu + x_{i,\ covar}^T \beta_{covar} + u_{strain[i]} + u_{b[i]} + \varepsilon_i \qquad \text{Equation 1}$$

where $\mu$ is the intercept, $\beta_{covar}$ are the fixed effects of covariates, $x_{i,\ covar}^T$ is the $i^{th}$ row of the covariate design matrix, $u_{strain[i]}$ is the effect of the strain of sample $i$, $u_{b[i]}$ is the effect the batch of sample $i$, and $\varepsilon_i$ is the error for sample $i$ with $\varepsilon_i \sim N(0,\ \sigma^2)$. The strain and batch effects were estimated as random effects: $\mathbf{u}_{strain} \sim N(0,\ I\tau_{Strain}^2)$ and $\mathbf{u}_b \sim N(0,\ I\tau_b^2)$. For the CC and founder strains, sex was included as a covariate. A similar model was fit for the DO mice but with no strain effect and diet was included as a covariate along with sex. The batch effects, estimated as best linear unbiased predictors (BLUPs) using restricted maximum likelihood estimates (REML),[77] were subtracted from each peptide measurement: $\tilde{\tilde{y}}_i^{pep\ k} = \tilde{y}_i^{pep\ k} - \widehat{u}_{b[i]}$.

For peptides expected to contain a polymorphism, we fit local genetic effects based on the haplotype at the marker closest to the TSS of the gene to which the peptide maps,

$$\tilde{\tilde{y}}_i^{pep\ k} = \mu + local[i] + x_{i,\ covar}^T \beta_{covar} + u_i^{kinship} + \varepsilon_i \qquad \text{Equation 2}$$

where $local_i$ is the effect of the local haplotype on peptide $k$ for sample $i$, $u_i^{kinship}$ represents a random kinship effect to account for overall genetic relatedness, and all other terms as previously defined. For the CC mice, $local[i] = p_i^T \beta_{local}$ where $p_i^T$ is the founder haplotype probability vector at the marker closest to the gene TSS (e.g., ordering the founder strains as AJ, B6, 129, NOD, NZO, CAST, PWK, and WSB, $p_i^T = [0\ 1\ 0\ 0\ 0\ 0\ 0\ 0]$ for a CC mouse $i$ that is B6/B6 at the locus). For the DO, $local[i] = d_i^T \beta_{local}$ where $d_i^T$ is the founder haplotype dosage vector, scaled to sum to zero, at the marker closest to the gene TSS (e.g., $d_i^T = [0.5\ 0.5\ 0\ 0\ 0\ 0\ 0\ 0]$ for a DO mouse $i$ that is AJ/B6 at the locus). For the founder strains, $local_i = x_{i,\ strain}^T \beta_{local}$ where $x_{i,\ strain}^T$ is the founder strain incidence vector for mouse $i$ (e.g., $x_{i,strain}^T = [0\ 1\ 0\ 0\ 0\ 0\ 0\ 0]$ for a B6 mouse). $\beta_{local}$ is an eight-element vector of founder haplotype effects, fit as a random effect: $\beta_{local} \sim N(0, I\tau_{local}^2)$ where $I$ is an 8×8 identity matrix and $\tau_{local}^2$ is the variance component underlying the local effects. The kinship effect is included for the CC and DO mice and modeled as $u^{kinship} \sim N(0, G\tau_G^2)$ where **G** is a realized genomic relationship matrix and $\tau_G^2$ is the variance component underlying the kinship effect, accounting for population structure.[78–81] Here we used a leave-one-chromosome-out (LOCO) **G**, in which markers from the chromosome the peptide is predicted to be located on are excluded from **G** estimation in order to avoid the kinship term absorbing some of $local[i]$.[82] We then calculated $r_{poly} = cor(\hat{\beta}_{local}, q)$, the Pearson correlation coefficient between $\hat{\beta}_{local}$, the BLUP of $\beta_{local}$ and $q$, the incidence vector of the B6 haplotype among the founder strains (e.g., $q = [01000000]$ for a peptide that contains a B6-specific allele that is missing in the other founder strains). Sets of peptides with polymorphisms were defined based on having $r_{poly} > 0.5$ for each of the CC, DO, and founder strains, to be excluded from further analysis because they would bias protein abundance estimation.

## Protein abundance estimation from peptides

Protein abundances were estimated from their component peptides after filtering out polymorphic peptides. The abundance for protein $j$ is calculated as $y_i^{prot\ j} = \frac{\sum_M y_i^{pep\ m} 1_{i,m}}{\theta_i}$ where $M$ is the set of peptides that map to protein $j$, $1_{i,m}$ is the indicator function that peptide $m$ was observed in mouse $i$, and $\theta_i$ is the scaling factor previously defined.[73] Similar to the previously described peptide normalization in the CC mice, proteins were scaled relative to the bridge sample and log-transformed: $\tilde{y}_i^{prot\ j} = log_2\left(\frac{y_i^{prot\ j} + 1}{y_{b[i]}^{prot\ j} + 1}\right)$.

For the DO and founder strains, there was no bridge sample, and proteins were instead normalized as: $\bar{y}_i^{prot\ j} = log_2(y_i^{prot\ j} + 1)$. Batch effects were removed from the protein data using the LMM described for the peptide data (Equation 1). If more than 50% of samples were missing a protein, it was removed from further analysis in order to avoid false downstream findings. Batch effects, estimated as BLUPs, were then removed: $\tilde{\tilde{y}}_i^{prot\ j} = \bar{y}_i^{prot\ j} - \hat{u}_{b[i]}$.

## Heritability estimation

We estimated heritability for all proteins in the CC, DO, and founder strains. The heritability model is similar to Equation 2, but for proteins instead of peptides and without the $local[i]$ term:

$$\tilde{\tilde{y}}_i^{prot\ j} = \mu + x_{i,\ covar}^T \beta_{covar} + u_i^{kinship} + \varepsilon_i \qquad \text{Equation 3}$$

where terms are as previously defined. The genomic relationship matrix **G** – corresponding to the kinship term $u^{kinship} \sim N(0, G\tau_G^2)$ for the CC and DO – is estimated from all markers, i.e., non-LOCO **G** – because there are no other genetic factors in the model. In the founder strains, $\mathbf{G} = X_{strain}X_{strain}^T$ where $X_{strain}$ is the founder strain incidence matrix. Sex was modeled as a covariate for all three populations, and diet as well in the DO. Heritability is then calculated as $h^2 = \frac{\tau_G^2}{\tau_G^2 + \sigma^2}$. The estimate in the DO is for the narrow sense heritability, representing the contributions of additive genetic effects. For the CC and founder strain mice, the estimate represents broad sense heritability, incorporating non-additive genetic effects, due to the presence of replicates.

## QTL analysis

In the CC and DO mice, we performed a genome-wide pQTL scan for each protein, testing a QTL effect at positions across the genome, using a model similar to Equation 2:

$$\tilde{\tilde{z}}_i^{prot\ j} = \mu + QTL_m[i] + x_{i,\ covar}^T \beta_{covar} + u_i^{kinship} + \varepsilon_i \qquad \text{Equation 4}$$

where $\tilde{\tilde{z}}_i^{prot\ j}$ is the standard normal quantile returned by the inverse cumulative distribution function of the normal distribution on the uniform percentiles defined by the ranks of $\tilde{\tilde{y}}^{prot\ j}$, i.e., the rank-based inverse normal transformation (RINT)[83] of protein $j$ for individual $i$, $QTL_m[i]$ is the effect of the putative QTL at marker $m$ on protein $j$ for individual $i$, equivalent to the $local[i]$ term in Equation 2 for the CC and DO mice, and all other terms as previously defined. The kinship effect was fit based on the LOCO **G** specific to the chromosome of marker $m$. We used RINT for the QTL analysis to reduce the influence of extreme observations that can produce false

positives, particularly when they coincide with a rare founder haplotype allele. This is of particular concern in the CC sample of 58 unique genomes. To test the QTL term, the model in Equation 4 is compared to a null model excluding $QTL_m$, summarized as the $\log_{10}$ likelihood ratio (LOD) score.

The QTL model in Equation 4 was also used for variant association mapping at specific pQTL identified through the haplotype-based analysis by adjusting the $QTL_m[i]$ term: $QTL_v[i] = p_{i,v}^T \beta_{QTL}$, where $p_{i,v}^T$ is the marginal variant allele probability vector for variant $v$, which is calculated by collapsing and simplifying the underlying founder haplotype probabilities based on variant genotypes in the founder strains (SQLite variant database: https://doi.org/10.6084/m9.figshare.5280229.v3).

For the CC mice, we mapped pQTLs based on strain averages where $\tilde{z}_i^{prot\ j}$ is the average of $\tilde{y}_{male,\ strain\ i}^{prot\ j}$ and $\tilde{y}_{female,\ strain\ i}^{prot\ j}$ followed by RINT across the strains. Founder haplotype probabilities were reconstructed at the level of individual mice and averaged for strain-level mapping. No covariates were included when mapping on strain averages. We tried mapping pQTLs in the CC mice on individual-level data, which returned largely consistent results, but notably fewer and weaker pQTLs. In the CC, we also mapped pQTLs to the mitochondrial genome and Y chromosome by testing whether the founder origin of the mitochondria or Y chromosome was associated with protein abundance. We fit Equation 4, treating the mitochondrial genome or Y chromosome as a single locus $QTL_Y[i]$ and $QTL_{MT}[i]$, respectively, using the non-LOCO **G** for the kinship effect. The founder strain of origin for the Y chromosome was determined for all CC strains. For the mitochondrial genome, six strains (CC031, CC032, CC041, CC051, CC059, CC072) possessed ambiguity between AJ and NOD, which we encoded as equal probabilities ($p_{i,MT}^T = [0.5\ 0\ 0\ 0.5\ 0\ 0\ 0\ 0]$).

### QTL significance thresholds

We estimated significance thresholds for pQTLs using permutations.[42] We accounted for missing data by performing 10,000 permutations of the normal quantiles for each level of observed missingness in the CC and DO mice (ranging from 0 to 50%). Genome scans of the permuted data used the model in Equation 4, excluding covariates and the kinship term. We first applied a genome-wide error rate correction across marker loci and then applied an FDR correction to account for testing multiple proteins.[43] We modeled the maximum (genome-wide) LOD scores from the permutation scans using a generalized extreme value distributions (GEV)[84,85] specific to each level of missingness, to compute genome-wide permutation $p$-value for each protein:

$$p_{perm}^{prot\ j} = 1 - F_{GEV,\ n_{NA}[prot\ j]}(max\ LOD[prot\ j]) \qquad \text{Equation 5}$$

where $F_{GEV,\ n_{NA}[prot\ j]}$ is the cumulative density function for the GEV fit from the permutations of quantiles with $n_{NA}$ number missing values, corresponding to the number missing for protein $j$, and $max\ LOD[prot\ j]$ is the maximum LOD score from the genome scan of protein $j$. We then used the Benjamini-Hochberg (BH) procedure[86] to calculate FDR $q$-values across the permutation $p$-values, and applied interpolation to find the permutation $p$-value that corresponds to FDR $< \alpha$: $p_{perm,\ \alpha}^{interp}$ where $\alpha \in [0.1, 0.5]$. Significance thresholds on the LOD scale, specific to FDR $< \alpha$ and $n_{NA}$ missing data points, were calculated: $\lambda_{FDR\ <\ \alpha}^{n_{NA}} = F_{GEV,\ n_{NA}}^{-1}(1 - p_{perm,\ FDR\ <\ \alpha}^{interp})$ where $F_{GEV,\ n_{NA}}^{-1}$ is the inverse cumulative density function for the GEV with $n_{NA}$ missing data points. As a final step to reduce random variation between sets of permutations, we regressed the estimated thresholds for a population and FDR level on the number of missing data points $n_{NA}$, and created a table of fitted thresholds: $\hat{\lambda}_{FDR\ <\ \alpha}^{n_{NA}}$ for $\alpha \in [0.1, 0.5]$ for both the CC and DO mice. Whether a pQTL met FDR $< \alpha$ significance, the threshold corresponding to $\alpha$ with the $n_{NA}$ for protein $j$ was used. For reference, $\hat{\lambda}_{FDR\ <\ 0.1}^{0} = 7.96$ and $\hat{\lambda}_{FDR\ <\ 0.5}^{0} = 6.33$ in the CC, and $\hat{\lambda}_{FDR\ <\ 0.1}^{0} = 7.86$ and $\hat{\lambda}_{FDR\ <\ 0.5}^{0} = 6.41$ in the DO.

### Defining local/distal status of QTL

Detected pQTLs were classified as local if their position was within 10 Mbp upstream or downstream of the middle of the coding gene. If they did not fall within this local window, they were classified as distal. The broad local window was used because the CC have larger LD blocks than the DO due to fewer outbreeding generations. With a narrower definition, it would be more likely to have "distal" pQTL in the CC that align and have consistent effects with "local" pQTL in the DO. On the other hand, this lenient definition of local may absorb some distally acting pQTLs that happen to be within 10 Mbp the gene on which they act.

### Consistency of QTL between the CC and DO

We evaluated the consistency of local and distal pQTLs between the CC and DO by comparing their haplotype effects. We first had to define pQTLs that were detected in both the CC and DO and thus pair them for effect comparison. Local pQTLs were paired based on simply having matching protein IDs. For distal pQTLs, we also required the pQTL positions to be within 10 Mbp of each other.

Haplotype effects were estimated at the pQTL marker using the model in Equation 4. To stabilize the effects, they were modeled as a random effect: $\beta_{QTL} \sim N(0, I\tau_{QTL}^2)$, where $\tau_{QTL}^2$ is a variance component underlying the haplotype effects of the pQTL. We then estimated the haplotype effects as BLUPs ($\hat{\beta}_{QTL}$). To declare pQTLs consistent between the CC and DO, we evaluated whether their haplotype effects were significantly positively correlated: $p_{QTL}^r = Pr(r_{QTL} > 0)$ where $r_{QTL} = cor(\tilde{\beta}_{QTL}^{CC}, \tilde{\beta}_{QTL}^{DO})$ and $r_{QTL}\sqrt{6}(1 - r_{QTL}^2)^{-1} \sim t_{(6)}$. To account for multiple testing, we used the BH procedure on the $p$-values for correlated effects and declared pQTLs with $q_{QTL}^r < 0.1$ as consistent between the CC and DO.

Haplotype effects for a pQTL are fit at a specific marker. Selecting which marker for effect comparison is complicated by the fact that the CC and DO have different sets of markers and the genomic coordinates of the peak LOD scores also vary. When comparing pQTLs detected in both populations, we fit the Equation 4 model at the markers with the highest LOD score specific to each population. When comparing pQTLs that were detected in only one population, we selected the marker in the population that failed to map the pQTL that was closest to the marker in the population that detected it.

### Consistency of local QTL in the CC with the founder strains

If the genetic effects on a protein are primarily local, the relative abundances for a protein in the founder strains should match the local pQTL effects observed in the CC and DO. We evaluated the consistency of local pQTLs in the CC with the founder strains, using an approach similar to how we compared pQTL effects between the CC and DO. For the founder strains, rather than fitting pQTL effects ($\tilde{\beta}_{QTL}$), we fit the founder effects as random terms (as described for the local term in Equation 2 for the founder strains) summarized as BLUPs ($\tilde{\beta}_{strain}^{Founders}$). We then calculated the Pearson correlation between local pQTL effects in the CC and founder effects in the founder strains: $r_{local} = cor(\tilde{\beta}_{QTL}^{CC}, \tilde{\beta}_{strain}^{Founders})$. As when comparing QTL effects between the CC and DO, we then tested $r_{local} > 0$, and corrected for multiple testing through the BH procedure.

### Mediation analysis

For each distal pQTL (lenient threshold) in the CC or DO populations, we performed a mediation analysis which involved a scan analogous to the QTL genome scans. Instead of scanning through genetic markers as putative QTLs, we scan through proteins as putative mediators of a given distal pQTL. The model is

$$\tilde{z}_i^{prot\ t} = \mu + QTL[i] + x_{i,\ covar}^T \beta_{covar} + mediator_q[i] + \varepsilon_i \qquad \text{Equation 6}$$

where $QTL[i]$ is as defined for $QTL_m[i]$ in Equation 4 but fixed at the peak marker $m$ of the distal pQTL for target protein $t$ and conditioned on protein $q$, with $mediator_q[i]$ representing its effect on protein $t$ for individual $i$, and all other terms as previously defined. The effect of the mediator is modeled as $mediator_q[i] = \beta_{prot\ q}\tilde{z}_i^{prot\ q}$, where $\beta_{prot\ q}$ is the regression coefficient for the mediator protein $q$ and $\tilde{z}_i^{prot\ q}$ is the RINT quantity of protein $q$ for individual $i$. The likelihood of Equation 6 model is compared to a null QTL model that excludes the $QTL_i$ term, producing a mediation conditional LOD score. The mediation model is fit for all proteins as individual mediators, excluding protein $t$, resulting in a mediation scan.

We assume that most of the proteins evaluated as candidates are not true mediators of the pQTL and thus the distribution of mediation conditional LOD scores approximates a null distribution, roughly centered around the LOD score of the distal pQTL that was first detected. We calculate the $z$-scores of the mediation conditional LOD scores and then define strong candidate mediators of the pQTL for protein $t$ as proteins with $z_q^{med} < -4$, where $z_q^{med}$ is the $z$-score of the mediation LOD score for candidate mediator protein $q$. The rationale being that when testing the $QTL$ term in Equation 6, if the mediator contains much of the information from the pQTL, its presence in both the alternative and null models will result in a large drop in the LOD score of the detected pQTL. For a protein to be declared as a candidate mediator of the distal pQTL, we required that the mediator TSS be within 10 Mbp of the pQTL marker. Strong mediators that were not near the pQTL often represent proteins that are correlated with the target protein $t$, which are often co-regulated members of a protein complex or pathway.

### Sex effects on protein abundance analysis

Proteins that exhibited differential abundance between the sexes, i.e., sex effects, were identified using an LMM similar to the heritability model (Equation 3) for the CC, DO, and founder strains, but instead testing the significance of the sex coefficient:

$$\tilde{y}_i^{prot\ j} = \mu + \beta_{Male}x_{i,\ Male} + x_{i,\ covar}^T \beta_{covar} + u_i^{kinship} + \varepsilon_i \qquad \text{Equation 7}$$

where $\beta_{Male}$ is the effect on protein $j$ of being male, $x_{i,\ Male}$ is an indicator variable of being male, and all other terms as defined previously. Other covariates and the specification of $u_i^{kinship}$ for the different populations are the same as described for heritability.

A $p$-value for the sex effect was calculated by comparing the model in Equation 7 to a null model without the sex effect through the likelihood ratio test (LRT): $p_{sex}^{prot\ j} = Pr(X > \hat{\chi}_{prot\ j}^2)$ where $Pr(.)$ denotes the $\chi_{(1)}^2$ probability density function and $\hat{\chi}_{prot\ j}^2$ is the observed LRT statistic for protein $j$. The LMM was fit with the qtl2 R package,[65] using maximum likelihood estimates (MLE) for parameters rather than REML, which are more appropriate for asymptotic-based significance testing of fixed effects. Proteins with significant sex effects were selected based on FDR < 0.1 using the BH procedure.[86]

We performed gene set enrichment analysis using the clusterProfiler R package.[63] We defined gene sets based on $q_{sex} < 0.01$ and split them further into subsets based on having higher abundance in males or higher abundance in females. We used the quantified proteins in each population as the background gene set. Hypergeometric tests for enrichment of GO and KEGG terms were performed with FDR multiple testing control.[87] Enriched GO and KEGG terms were selected based on having $q_{set} < 0.1$.

**Protein complex analysis**

We assigned proteins to protein complexes using annotations.[49] For each protein complex, we quantified how tightly co-abundant, i.e., cohesive, the members are, by calculating the median pairwise Pearson correlation for each protein with the other members of the complex. We summarized cohesiveness within a complex by recording the median and interquartile range across the median correlations for the individual proteins.

To assess whether genetic factors or sex regulated protein complexes, we estimated the complex heritability and complex sex effect size based on the PC1 from PCA[51] of the abundances of the proteins annotated to the complex. We first filtered out proteins with local pQTLs (FDR < 0.5) or strong distal pQTLs (FDR < 0.1) to minimize the influence of proteins with independent genetic effects in order to focus on the shared effects on a protein complex. We also regressed out effects of covariates from the individual proteins prior to PCA in order to keep the PC1 summary from reflecting their effects. To estimate complex heritability, we removed the effect of sex in the CC, and both sex and diet in the DO. For complex sex effect size, we removed the effect of diet from the DO. We estimated complex heritability using the model in Equation 3, with no covariates and the complex PC1 as the response variable.

To estimate the complex sex effect size: $\phi_{sex}^2 = 1 - \left( \sum_i e_i^2 \Big| M_A \right) / \left( \sum_i e_i^2 \Big| M_0 \right)$ where $\sum_i e_i^2 \mid M_A$ is the sum of squared residuals (SSR) under the alternative model (Equation 7) and $\sum_i e_i^2 \mid M_0$ is the SSR under the null model (Equation 7 excluding sex effect). Interval estimates for complex heritability and complex sex effects represent 95% subsample intervals. We randomly sampled without replacement 80% of the CC and DO data 1,000 times and estimated the complex heritability and complex sex effects for each subsample as well as the 2.5th and 97.5th quantiles across the subsamples. We estimated summaries for protein complexes that had four or more proteins observed in the CC or DO, after removing proteins with local pQTLs (FDR < 0.5) or distal pQTLs (FDR < 0.1), thus limiting the potential that the PC1 reflected a strong pQTL not shared by other members of the complex.

**Strain-specific outlier proteins**

To identify proteins with low or high abundance characteristic to individual CC strains, we fit the following LMM:

$$\tilde{y}_i^{prot\ j} = \mu + \beta_{Male}x_{i,\ Male} + u_{strain[i]} + \varepsilon_i \qquad \text{Equation 8}$$

with all terms as previously defined. Effects for all CC strains for each protein $j$ $(\hat{u}_{strain}^{prot\ j})$ were estimated as BLUPs, which were then transformed to z-scores per protein $(\mathbf{z}_{strain}^{prot\ j})$. We defined a strain-specific protein outlier to be a protein $j$ in CC strain $i$ for which $\left| z_{strain\ i}^{prot\ j} \right| > 2.5$. This represents a lenient threshold because we aim to cast a wide net and identify interesting characteristics of CC strains, potentially due to subtle effects across many proteins. We intersected the strain outliers with known CC strain-specific genetic variants based on CC strain identity and the annotated coding gene,[30] identifying variants that likely have local effects on protein abundance.

For each CC strain $i$, we defined sets of proteins that had consistently low, high, and extreme (low or high) abundance based on their strain effects: $\Omega_{strain\ i}^{high} = \{prot\ j: z_{strain\ i}^{prot\ j} > 2.5\} \ \forall\ j$, $\Omega_{strain\ i}^{low} = \{prot\ j: z_{strain\ i}^{prot\ j} < -2.5\} \ \forall\ j$, and $\Omega_{strain\ i}^{extreme} = \{prot\ j: \left| z_{strain\ i}^{prot\ j} \right| > 2.5\} \ \forall\ j$, respectively. We then tested whether the CC strain-specific outlying proteins were enriched in GO and KEGG terms ($q_{set} < 0.1$).

**ADDITIONAL RESOURCES**

All processed data and pQTL results are also available for download and interactive analysis from the QTL Viewer webtool (https://github.com/churchill-lab/qtlapi) for both the CC (https://qtlviewer.jax.org/viewer/FerrisCC) and DO (https://qtlviewer.jax.org/viewer/SvensonHFD). All these resources are also listed in the key resources table.

