## [Document S2. Transparent peer review records for Keele et al. · Cell Genomics]

Cell Genomics Transparent Peer Review Report

Title: Regulation of protein abundance in genetically diverse mouse populations

Corresponding Author: Gary A. Churchill

Editor: Orli G. Bahcall

Review: 3 referees, 2 rounds of review

Editorial summary of report:

Gary Churchill and colleagues extend report an exciting resource of mass-spectrometry profiling of liver tissue from male and female mice across 116 Collaborative Cross (CC) mice across 58 inbred strains, extending the utility of this foundational resource of genetically diverse mouse populations to enable analyses of the regulation of proteome. The authors analyze the genetics of protein abundance in the CC, Diversity Outbred (DO) mice and their founder strains. They map protein abundance quantitative trait loci (pQTL) between the CC and DO for individual proteins and for protein complexes, and find that the genetics of individual protein regulation is highly conserved across the mouse populations. They demonstrate the consistency of mass-spec proteomics data across experiments and conservation of the genetic regulation of protein abundance across these resource populations.

The manuscript was sent for two rounds of peer review by 3 referees, with referee reports and author responses as included in this file. The editor requested revisions at each stage to respond to the technical queries of the referees and to improve the reporting, presentation and discussion of the work.

First Review, Referee Reports:

Referee #1

In this paper, Keele et al present a pQTL mapping study in the mouse Collaborative Cross and use the data for an in-depth comparison to their earlier pQTL mapping in mouse Diversity Outbred animals. The study is technically very well done, and the paper is clearly written. The figures are informative and beautifully made.

The results show extensive sharing of genetic signals between the CC and DO panels, at the levels of heritability, sex differences, and pQTLs. This result is reassuring as it suggests that both the CC and DO datasets are of high quality. At the same time, the high concordance between the datasets does limit novelty to some extent. Further, the (excellent) experimental and analysis methods were all established earlier by the authors, such that the current manuscript does not provide major advances in these areas.

To compensate for this, the authors perform a deep dive into the nature of genetic effects on protein complexes, which does expand significantly on earlier work. I enjoyed reading the careful dissections of the various complexes and the range of genetic architectures that affect them, ranging from strong, essentially Mendelian effects caused by variants introduced by a single founder strain to more polygenic architectures. While also not entirely conceptually novel, these sections make the paper into a solid advance.

Specific comments:

1. Page 26 (mediation methods): "The likelihood of Equation 5 is compared to a null QTL model that excludes the QTL_i term, producing a mediation LOD score." I don't follow this logic and suspect that there is something missing from this description. Specifically, if the trans-pQTL truly acts via the mediator, the presence or absence of the QTL marker in the model should not matter because the influence of the QTL is contained in the abundance of the mediating protein. Therefore, comparisons of the two models explained in the text (that both include the mediator) should not produce a high LOD score *especially* if there is mediation. This seems like the opposite of the intended outcome.

Do the authors compare the QTL effects between two separate model comparisons: 1. Effect of the QTL in a model without the mediator; this is simply the QTL mapping model, versus 2. Effect of the QTL in a model with the mediator?

It would be helpful to clarify the explanation of the mediation testing procedure.

2. The introduction ignores important pQTL citations in all species the authors consider (not to mention those in other species, such as plants). In yeast, the Kruglyak lab has conducted multiple pQTL analyses over the years. In humans, there is earlier pQTL work from the Snyder lab (<https://www.nature.com/articles/nature12223>). In mice, there are pioneering pQTL studies from the Lusic lab

(<https://journals.plos.org/plosgenetics/article?id=10.1371/journal.pgen.1001393>). A more comprehensive overview of the pQTL literature would help place the current results in context.

3. Likewise in the introduction, the treatment of local versus distant pQTLs is too simplistic. Perhaps due to brevity, the first paragraph creates the impression that there is a consensus in the field that local pQTLs are mostly due to underlying mRNA variation, while distant pQTLs are mostly post-translational. No such consensus exists. For example, there are many instances of protein-specific local pQTLs in the literature (for example in the Battle 2015 paper), as well as thousands of distant eQTLs that affect mRNA levels, mostly with unknown effects on the corresponding proteins. I'd like to see more nuance in the authors' treatment of the pQTL literature in the introduction.

4. Please add a brief justification for using liver in this and the previous (in the DO mice) study.

5. Page 4, and the corresponding sections in the Discussion on mediation: it is true that mediation analysis cannot detect unobserved mediators that are not measured in the data, and that it can instead flag false mediators that are linked to such factors (as the authors explain for Naxd). In addition, trans-acting variants need not act via abundance variation of the causal factor at all, if they alter the activity/function of the factor by changing its protein sequence. For example, a nonsynonymous variant that alters the activity (but not the abundance) of a transcription factor could affect gene expression in trans and would be undetectable by mediation analysis. This possibility should be acknowledged as another possible source of false negatives and false positives in mediation analysis.

Minor comments on presentation:

6. Line numbers would have been helpful to this reviewer.

7. Page 3 top: At 10 Mb, the criterion for calling a pQTL "local" will span multiple genes and therefore could technically sometimes include trans-acting pQTLs that happen to be located close to the given gene. I assume this wide window is based on the mapping resolution in the linkage data (as opposed to GWAS in an outbred population). It would be helpful to state this explicitly, and add the width of a typical pQTL in the data for context.

8. End of intro: "[...] became fixed during the of the CC" - something is missing here.

9. Page 4 "cannot be ruled if they unobserved" misses an "are"

10. Page 4 end of main paragraph: ", which revealed similar levels of concordance [...]": It is not clear from the sentence what these levels of concordance are similar to. I suggest rewording this sentence, as it currently obscures a key result from this section (that mediation signals are shared between CC and DO).

11. Page 5 bottom: "PC1" was not defined in the main text before this use.

12. Figure 2A: y-axis label (and the equivalent information in the legend) should read "gene position". Proteins do not have positions in the genome; the genes that encode them do. In addition, a scale for dot size (i.e., LOD score) would be helpful.

13. Figure 2J: the legend text for the circus plot (i.e. panel J) runs seamlessly into that for panels K & I. Please delineate these panels more clearly.

14. Figure 2 K & I legend: explain what the LOD score of 12 (dashed horizontal line) denotes. Genome wide significance? FDR = 0.05? The same comment applies to all other plots of this type (e.g. those in Figure 4).

15. Related, why are the lines in the LOD profiles in Figure 4 (and other figures later on) plotted at LOD=6 rather than LOD=12 like in Figure 2?

16. Figure 4 legend: "observance of the homozygous PWK genotype": "observance" seems an odd choice of words here. Maybe "observation" or "presence" is better? Also, in "distally control other members" is should read "controls". Finally, panel "h" is incorrectly called "e" in the legend.

17. Figure 6k legend: "The relationship is negative" should be deleted.

Referee #2

Reviewer comments for the Cell Genomics article "Regulation of protein abundance in genetically diverse mouse populations" by Keele et al.

The authors have acquired and analyzed liver proteomics data for a fairly large large population—116 individuals—and diverse inbred mouse population and they have performed a meta-analysis of these results against the results of a conceptually similar previous 2016 study in a distinct (though related) mouse population of about 220 individuals. The primary difference between the two studies is that this study is entirely on inbred strains, but more or less it is the same idea and makes an ideal case study for population meta-analysis of proteomics data, which to my knowledge has not been done before. This study thus provides a useful baseline for proteomic variation. I have a number of concerns, but all of which should be relatively easily either rebutted, or addressed in a minor revision.

Scientific comments, in approximate order of location in the paper

- (1) The authors mention in Fig 2D how many local and distal pQTL they identify, with around 25% higher QTL detections in the DO population for the 4556 proteins quantified in both. The DO population is both larger in sample terms and more genetically variable than the CC so this is not especially surprising in one way, but the lower observed heritability would flip in the other direction and favor CCs. What's the relative contribution of sample size, variability, etc?
- (2) I like how you have added the lenient detection comparison (Fig S1F). One related question: for those ~14% of QTLs that are dramatically different between CC and DO, do these fall into any patterns? i.e. genes in large complexes, mitochondrial genes, genes which have fewer sequence variants, ...? I just very quickly checked the top 30 negative correlates in DAVID and did not see any enriched ontologies.
- (3) The authors do mention this, e.g. "COPS8 is the strongest mediator for ..." (page 4) and mention that it "may be less accurately measured in [their] DO sample population". Why would that be the case? I know the LC-MS setup is different (question on that later), but how different is this? E.g. how many peptides are measured for COPS8? How many NAs/0s does its average peptide have? Are QTLs—both cis and trans—that disappear in CC and appear in DO or vice versa, are these more likely to be protein measurements that are the synthesis of only one or two peptides?
- (4) Page 5: "GSEA revealed that proteins related to ribosomes, translation, ... were more abundant in male livers, while ... were more abundant in female livers". Is this expected? This seems surprising to me and I don't think I've ever seen that in literature. Why would ribosomes, translation, etc, be higher in male livers? Certainly not impossible, but I haven't come across it and there is no reference cited, so if the authors are confident in this data, I think they should visualize it in a figure. That the directionality is the same in the DO and CC is certainly a very good clue that it is a real effect, so I don't doubt it, I would just like to see it as it is novel as far as I know and nothing especially relevant came up in a cursory literature search.
- (5) Page 6: "Notably, complex-heritability is consistently higher in the CC than DO and uncorrelated with complex-heritability in the DO". First off, could you directly put the "CC" and "DO labels directly on Figure 3 and Fig S4A/B so readers don't have to go to the legend to see

which is red and which is blue? (I know you do it consistently across figures—thanks for that—and you almost always list CC and DO in the figure panel itself, so just a small request here for people like me who quickly forget the color coding). Second off: This shift in h^2 for complexes between DO and CC looks like exactly the same value as the shift between non-complexes between DO and CC, so couldn't you take any subset of genes and make the same statement? It seems like it's just a uniform downward shift in heritability for all genes in DO compared to CC. The decrease in heritability in complex members compared to non-complex members is very strong and robust, so I have no issue with the overall message. The decrease in heritability is also expected; complex proteins also tend to have far fewer cis-pQTLs compared to an average gene, presumably because PPIs are disproportionately important to regulating their expression. I have no real issue with the text here, it's just not particularly novel by this point.

(6) Related to the previous two points: "Protein-complexes previously shown to be driven by sex, such as eIF2B were confirmed...". The authors mentioned in the previous section that ribosomes were more abundant in males, but I don't see a sex effect here; mitochondrial ribosome large and small subunits seem to have no sex effect, and the ER ribosome is not highlighted. I know the previous section was about GSEA and this is about heritability and correlation—but shouldn't it show up in both?

(7) The proteasome difference is fairly impressive: that's quite a few subunits with very clear changes. Are there any known phenotypes?

(8) The mention of "notable exception is Auh". It's interesting and a nice hit, but I'm not sure it's a "notable exception". I found one paper referencing it "associating with" the mitochondrial ribosome, but I wouldn't really call that a complex member in the normal sense of the word. Otherwise, I don't see Auh considered a canonical component of the mitochondrial ribosome in literature, it's not in CORUM as a mito ribosome component, and it is also not really picked up by BioPlex nor STRING. STRING even puts it with cholesterol biosynthesis (although I am always a bit skeptical of STRING). I also don't see AUH on GO's list of "mitochondrial translation". Similar question: why is RPS15 included in this figure? Also some other MRPS genes are measured, but are missing, e.g. MRPS27 and 28 are referenced in the supplemental tables as measured in the CC, but they're not here. PPME1, same question. METTL17 makes sense and fits in both better here in the data, and in BioPlex although not STRING.

(9) 6046 strain-protein outliers for 4323 proteins across 58 CC strains sounds like far too many to be accounted for by private spontaneous mutations. Shifman 2006 (PMID 17105354) only detected a few dozen SNPs in far older RI strain populations. I didn't immediately find a number from the 2019 Shorter paper but in any case it should be quite rare. I would expect the vast majority of those 4323 outlier proteins to be noise brought on by measurement issues or normalization issues. How many of those outliers are related to proteins that are measured by a single peptide, for instance? The examples shown are convincing and interesting, but they map to known private variants. The authors do mention this ("measurement error in proteomics is likely from [among other things] the number of peptides used to summarize it...").

(10) The authors mention later that CC013 has a unique liver phenotype and show it in Figure 7—that's neat! But then they next mention "altered complex I function in C007" but do not show or reference it. Did I miss that somewhere? I don't see anything mentioned on its strain page (<https://www.jax.org/strain/029625>).

(11) Methods: The new CC data were acquired on an Orbitrap Fusion Lumos using a different column with different parameters (beads, etc) compared to the DO data from 2016, acquired on an Orbitrap Fusion Tribrid. Also I may have missed it, but I don't see how long the acquisitions were in this study (36 hours per 10-plex in DO). I'm kind of surprised that the Lumos does not detect more proteins than the Tribrid (both are around 6800). How many individual unique peptides were measured in the two? Are the measurements more stable in the CC dataset? Could this account for the improvement in observed heritability (Figure 1C) in the CC compared to the founders? While generally I think the authors' findings match what I would expect, I am worried slightly that some of the meta-analysis is an "oranges to clementines" comparison. Of course I am glad the authors used the newer machine to acquire this data, but it makes it a bit trickier to be certain that differences between CC and DO are due to the population compared to the acquisition. There are probably many ways to do approximations of the effect size of this potential issue. Ideally it would have been great to re-run some of the exact same peptide samples from the DO study that had been sitting in a freezer for 4 years, but in lieu of that, there are a few other ways to estimate differences in data quality. I have thought about this for a few minutes and I always come up with caveats. My general expectation would be that the ground truth of observed heritability for CC vs CC Founders should be equivalent. The increased in observed heritability for CC makes me wonder if it's due to a newer machine, or it could just be the increased number of strains raises the observed heritability (I guess this latter hypothesis is easily testable by bootstrapping, or may indeed be already known in literature as it sounds like a straightforward idea to check).

Non-scientific comments:

- (1) Typo on page 4: "ruled out if they unobserved" (missing "are") and "BCKDHB for Bckdha and Ppm1k for," (Ppm1k for what?).
- (2) Typo page 11: "exosome with a single large effect pQTL local".
- (3) The figure legends are long. Like VERY long. It is four full pages of figure legends for the main story, versus eight pages for the entire results section. I like how the results were concise and to the point for such a complex paper, but the figure legends need to be trimmed. There is a lot more explanation in the legends than needed.
- (4) I downloaded the full 10 GB dataset and found it surprisingly easy to load the data and work with it. I didn't do much with it besides check that I can see it, but at least everything seems to work. The final output data in phenotypes/debatched is a little unintuitive for someone without a basic bioinformatics background to work with. Maybe this is a good impetus for them to learn how to use it, but it'd be nice if it was in a format where anyone could immediately see "does gene X correlate with gene Y in their data". Right now they'd have to do a bit of parsing and quite a bit of lookups to convert proteinIDs to gene symbols. Maybe this is fine, as if someone can't do this then it would be difficult to really analyze the data properly. For systems biologists it is certainly usable in its current state, but I think it is still slightly beyond the basic Excel skills of your average molecular biologist.

The authors have generated a wealth of liver proteomics data in 116 individuals from 58 collaborative cross mouse lines and have performed a detailed meta-analysis of this study

compared to their earlier and conceptually-similar 2016 work. These results add to the growing, but still quite small, body of literature detailing how large-scale proteomics data can be used to study genetic interactions, and it highlights both a few broad patterns in proteome regulation as well as identifying a select few interesting specific hypothesis either for future examination, or for which they have done their own pilot study here, as for CC013. I have some concerns that the broad analyses comparing DO to CC may be a "clementines to oranges" type comparison due to differences in the acquisition technology which are not addressed: proteomics is changing fast enough still such that a dataset generated 4 years later on the next-gen of LC-MS might well bias the data to be improved in the new dataset regardless of the ground truth. In many cases, the meta-analysis results are so clear-cut that this issue cannot possibly affect the results (i.e. very high cis-pQTL overlap, very low trans-pQTL overlap) but in other cases it is potentially an issue, such as the shift in observed heritability for DO vs CC: which is certainly what one would expect, but may be further biased by the analysis. I have two much more minor concerns, one about the way that complexes were selected (especially for the mitochondrial ribosome) and about making the data trivial for someone to analyze on their own (rather than its current "reasonably easy" state). Converting the *_proteins_data.csv files to an easily-used format would take only maybe a couple hours for someone with basic bioinformatics background to do, but it is certainly not possible in Excel, which unfortunately remains a mainstay, especially for molecular biologists who may be interested in the dataset, but only if it is utterly trivial to analyze. For systems biologists, it is quite fine as it is. Generally, it would be nice to not have to download the entire 10 GB package to just get at the peptide and protein data, even though I enormously appreciate the good organization, reasonably clean code, and completeness of their zip file (e.g. if it does not exceed size regulations, put it as a supplemental table).

I have written an extremely long review because I think this paper is likely to become an excellent reference for the next 5 years going forward as proteomics datasets become increasingly mainstream, and as analysis (and meta-analysis) of proteomics datasets is becoming an increasingly-realistic possibility for people who are not proteomics specialists themselves.

One concern: the take-home message is probably not so clear for those who don't do mouse/population genetics or proteomics. There are several take-home messages that jump to mind; for me I think it's perhaps more interesting (and welcome) that the significant findings from proteomics are so congruent between two distinct and independent populations. That is: significant findings that people pick out from routine proteomics are also likely to be reliable. The abstract seems like it could use some more force to it that will excite an average molecular biologist to say "hey, we should use this dataset" or "hey, we should use proteomics". Right now, I think it only says that point to molecular biologists who are studying large complexes.

Referee #3

The MS has been submitted to Cell Genomics, a brand-new journal, so it is difficult to assess its specific suitability but it seems to be within the journal's remit. Assessed on its own terms, the study represents a massive amount of work and is scientifically rigorous, with some interesting

findings. My overall impression is that rather too much material has been compressed into a single paper. I assume the study has been submitted as a full paper, where the maximum length of the full text is 45000 characters, which is about 15-16 single spaced pages and roughly the length of the MS main text as submitted (I can't count the number of characters in a PDF). Despite being about maximum length, there is still a great deal of material in the supplement or methods which is barely touched on in the main text. Some of the figures in the main text (again at the maximum) contain far too many sub-figures (eg Fig 6 has 20 sub-figures). 5/7 of the main figures are concerned with protein complexes, but only about half the results text. I don't think this makes for a well-balanced paper.

The main scientific result is that the genetic architecture of the mouse proteome is largely conserved in two populations - one inbred (the CC: data new to this study) and one outbred (the DO: data from a previous study [Chick et al by the same lab]) - both descended from the same inbred founders, and therefore with the same pool of segregating variation. The study shows that cis-acting pQTLs mostly replicate but trans-pQTL do not, but I think this is likely because the small sample sizes in the study do not have enough power to map trans effects reliably.

This finding, whilst not surprising, is important and is well demonstrated, although I have some comments below regarding the analysis performed, particularly regarding non-additive genetic effects. It might be worth contrasting the result with that found in *Drosophila* by Trudy Mackay's group, where very different genetic architectures (but for non-proteome traits) were found in two populations related in a similar way.

My main criticism is that apart from this discovery, the rest of the paper on the protein complexes is more of a descriptive study, albeit often interesting, but without any particular hypotheses or questions to drive the analysis, and therefore rather vague conclusions. This is reflected in the abstract which is also rather tentative. The story arc for the protein complexes is a negative one: that with certain interesting exceptions genetics does not drive complex abundance consistently between DO and CC populations, and that instead the cohesiveness of a complex is its key measure, presumably under environmental or temporal control, and with stoichiometric feedback to ensure the relative abundances of a complex's constituent proteins are matched.

Once genetics or sex stops being the organising principle, the MS becomes more a study of special cases, each individually very interesting and with its own figure (such as exosomes, Fig 4, and chaperonins, Fig 5, and proteosomes Fig 6, but I am not sure Fig 7 is worth including) but without a clear take home message. Perhaps some of these stories should be expanded into stand-alone papers in more specialist journals? Each of these mini-studies is very impressive and deserves fuller exposition. But I needed to examine the figures at very high magnification and spend a lot of time figuring out what everything meant to appreciate the stories. It did not help that key interpretations were buried in the figure legends rather than in the main text. This is a general point affecting all the figures and should be addressed. Sometimes there are

inadequate descriptions of the figures and one has to guess exactly what some of the figures mean.

One other point regarding the protein complexes. The analyses often use PC1 as a one-dimensional summary statistic (effectively an eigengene). This might not necessarily be the most relevant statistic, because PC1 finds the weighted combination of constituent proteins for a complex with the largest possible variance between individuals. This is tuned to identify situations where the constituents lack cohesiveness, for example where the 26S proteasome exists in two forms in different CC lines. Thus genetic mapping using PC1 would potentially reveal loci associated with variation in the complex and similarly for the heritability. On the other hand the average of the standardised abundances of the constituents would give a measure of the overall abundance of the complex, and this might have different genetic drivers and a different heritability. Did the authors map QTLs for average protein abundance?

Overall this is an impressive study but needs reworking to make it clearer and cleaner, for example by pruning the number of figures. I found it a very difficult paper to review because it resembles an iceberg, where only a fraction of the tremendous effort that has gone into it is accessible without diving in very deeply.

Detailed Comments:

Some of these questions are simply requests for clarification, others require some work.

(a) Genetic architecture:

(i) I suggest simultaneously estimating additive and dominance h^2 in the DO using two GRMs (GCTA lets you do this, for example). This may have bearing on the differences in protein complex heritability discussed on page 6.

(ii) In Fig1d-f, please add the linear regression lines of (say) CC heritability on DO heritability etc

(iii) Why not do a simultaneous analysis of CC and DO so that you can test formally if the haplotype effects are the same? That is, fit a model like $y \sim \text{genotype} * \text{population}$, where genotype is either a SNP dosage or haplotype dosage, and population is a factor indicating CC or DO.

(iv) Define stringent and lenient thresholds in the main text

(v) In the methods both haplotype- and SN based QTL mapping are described but (I think) only haplotype mapping is used in the results. Is that correct? If so, do the Methods even need to consider SNP-based mapping?

(vi) In the Methods, both peptide and protein phenotypes are defined, but only protein phenotypes are reported in the Results, correct?

Fig 2k,l: What are the bands of pale grey dots on the figures? How do the diagrams under each Manhattan plot relate?

(b) Mediation analysis: The Naxd story is unclear and needs more explanation

(c) Sex effects.

(i) Fig 1g,i suggests that DO sex effects are slightly larger on average than CC or founders. Please work out the linear regression lines and test if their slopes are different from 1. (ii) A few CC strains are known to have females and males of similar weight. Do these strains show the same level of sex difference as strains where the weights are different? In other words, is there a correlation between sex difference in weight and average protein sex difference, across the CC strains?

(d) Protein complexes:

(i) Define complex cohesiveness and complex heritability in the Results, not just the Methods, and explain their significance.

(ii) I note cohesiveness is defined as the median pairwise correlation of proteins in a complex. What happens with negative correlations? What would happen if it was defined as the minimum absolute correlation?

(iii) See my comment above relating to PC1

(e) Figures

Fig3a,c,e -Presumably the color codes are the same as in other figures (orange=CC, pale blue=DO) . Please indicate on the figure.

Fig 4 Exosome

Fig 4 d,e,g,h What is the reason for sometimes plotting sex comparisons as scatter plots or as `geom_jitter()` plots? Surely it would help clarity to use a consistent plot design. A similar point applies to Fig 5g,h

Fig 5 (Chaperonin complex)

If have understood the genetic architecture correctly, in DO the genetic control of the Chaperonin complex is driven by a trans pQTL for CCT4 on chr5 that is recessive for NOD. In CC mice it is driven by a combination of the same trans pQTL plus a cis pQTL on chr11 due to the PWK allele.

(i) Fig 5a,b what are the grey dots? Are the p-values for these Manhattan plots well calibrated (as shown by a qq-plot).

(ii) What is the evidence that the chr5 QTL is recessive? The fig 5c presumably shows the DO hets as shades of gray (this is not explained in the legend but there is a haplotype dosage greyscale below Fig 5a possibly related to this) which might support a recessive effect but it would be nice to see a formal statistical test.

(iii) if 5f, do dashed lines of any color mean LOD <6 in CC?

Fig 6.

(i) What is the point of Figs 6e,f? They seem to suggest nothing beyond the Manhattan plots 6g-j. Minor point - the length scaling of the chromosomes on 6e,f is different from that in 6g-j so the chromosomes don't align.

(ii) Fig 6p-s: I don't understand the story being told in these figures. I also don't understand the colors in 6p,q (why is one black, one brown?) And what do Fig 6k, l n, o tell us?

Fig4, Fig5, Fig 6: Some of the plot types are the same between the figures, but differ in small ways. eg. 4h 5h use different coloring rules. It would help to be completely consistent across figures.

(f) Discussion:

Not sure I follow this argument:

"comparing the CC to the DO reveals the impact of an inbred genetic background on a number of protein- complexes, due to recessive effects, or conversely, the lack of dominance genome-wide. Furthermore, CC strain replicates can capture multi-locus interactions, i.e., epistasis, by fixing alleles at multiple loci within a strain."

Why is the CC better for dissecting epistasis than an outbred population like the DO?. Also I am not sure why recessive or dominant effects are easier to detect in the CC, if that is what is being claimed. In fact the reverse is surely the case, because inbred populations cannot distinguish additive from non-additive genetic effects - DO heterozygote genotypes are the only sources of information that can make this distinction.

Response to reviewer comments for
“Regulation of protein abundance in genetically diverse mouse populations”
[CELL-GENOMICS-D-20-00070]

Reviewer #1 summary:

In this paper, Keele et al present a pQTL mapping study in the mouse Collaborative Cross and use the data for an in-depth comparison to their earlier pQTL mapping in mouse Diversity Outbred animals. The study is technically very well done, and the paper is clearly written. The figures are informative and beautifully made.

The results show extensive sharing of genetic signals between the CC and DO panels, at the levels of heritability, sex differences, and pQTLs. This result is reassuring as it suggests that both the CC and DO datasets are of high quality. At the same time, the high concordance between the datasets does limit novelty to some extent. Further, the (excellent) experimental and analysis methods were all established earlier by the authors, such that the current manuscript does not provide major advances in these areas.

To compensate for this, the authors perform a deep dive into the nature of genetic effects on protein complexes, which does expand significantly on earlier work. I enjoyed reading the careful dissections of the various complexes and the range of genetic architectures that affect them, ranging from strong, essentially Mendelian effects caused by variants introduced by a single founder strain to more polygenic architectures. While also not entirely conceptually novel, these sections make the paper into a solid advance.

Thank you for the positive summary.

The methods described under sections “Filtering out peptides that contain polymorphisms” and “Protein abundance estimation from peptides” are substantially new and are described for the first time here.

Specific comment #1: Page 26 (mediation methods): "The likelihood of Equation 5 is compared to a null QTL model that excludes the QTL term, producing a mediation LOD score." I don't follow this logic and suspect that there is something missing from this description. Specifically, if the trans-pQTL truly acts via the mediator, the presence or absence of the QTL marker in the model should not matter because the influence of the QTL is contained in the abundance of the mediating protein. Therefore, comparisons of the two models explained in the text (that both include the mediator) should not produce a high LOD score *especially* if there is mediation. This seems like the opposite of the intended outcome.

Do the authors compare the QTL effects between two separate model comparisons: 1. Effect of the QTL in a model without the mediator; this is simply the QTL mapping model, versus 2. Effect of the QTL in a model with the mediator?

It would be helpful to clarify the explanation of the mediation testing procedure.

We agree that the description of mediation analysis was inadequate and so have updated it (**lines 908-916**). Reviewer #1's interpretation is correct. The disconnect is due to the fact that, based on our approach, a strong mediation signal does not correspond to a high mediation LOD score, but rather a low mediation LOD score. The updated text describes an example from the results (**Figure 2j**) in detail to highlight the rationale. We also note that there were two Equation 5's and have corrected it.

Specific comment #2: The introduction ignores important pQTL citations in all species the authors consider (not to mention those in other species, such as plants). In yeast, the Kruglyak lab has conducted multiple pQTL analyses over the years. In humans, there is earlier pQTL work from the Snyder lab (<https://www.nature.com/articles/nature12223>). In mice, there are pioneering pQTL studies from the Lusk lab (<https://journals.plos.org/plosgenetics/article?id=10.1371/journal.pgen.1001393>). A more comprehensive overview of the pQTL literature would help place the current results in context.

Thank you - the omissions were an oversight on our part, and we have updated the background pQTL section of the Introduction. Specifically, we added references from human data (Snyder lab), mouse (Lusis lab), yeast (Kruglyak lab), and *Arabidopsis* (Jansen lab) (**lines 45-49**).

Specific comment #3: Likewise in the introduction, the treatment of local versus distant pQTLs is too simplistic. Perhaps due to brevity, the first paragraph creates the impression that there is a consensus in the field that local pQTLs are mostly due to underlying mRNA variation, while distant pQTLs are mostly post-translational. No such consensus exists. For example, there are many instances of protein-specific local pQTLs in the literature (for example in the Battle 2015 paper), as well as thousands of distant eQTLs that affect mRNA levels, mostly with unknown effects on the corresponding proteins. I'd like to see more nuance in the authors' treatment of the pQTL literature in the introduction.

We agree that the description of local versus distal pQTL is brief and we skipped over some of the nuances for brevity. We based our assertion on Chick et al (<https://doi.org/10.1038/nature18270>) where we showed that in the DO mice a majority of local pQTL are mediated by their transcript. We have adjusted the text to acknowledge a wider range of causes of local and distant genetic effects (**lines 53-55**).

Specific comment #4: Please add a brief justification for using liver in this and the previous (in the DO mice) study.

The original DO study investigated the effects of standard chow versus high fat diets; see Svenson et al (<https://doi.org/10.1534/genetics.111.132597>) and Gatti et al (<https://doi.org/10.1101/098657>). Liver was chosen because of its central role in metabolism. The CC cohort was collected later and raised on standard chow diet. Other tissues have been collected from these same CC mice, but we first analyzed liver to match the DO samples. We have added text to the Methods describing why liver was used (**lines 606-607**).

Specific comment #5: Page 4, and the corresponding sections in the Discussion on mediation: it is true that mediation analysis cannot detect unobserved mediators that are not measured in the data, and that it can instead flag false mediators that are linked to such factors (as the authors explain for Naxd). In addition, trans-acting variants need not act via abundance variation of the causal factor at all, if they alter the activity/function of the factor by changing its protein sequence. For example, a nonsynonymous variant that alters the activity (but not the abundance) of a transcription factor could affect gene expression in trans and would be undetectable by mediation analysis. This possibility should be acknowledged as another possible source of false negatives and false positives in mediation analysis.

Reviewer #1 is correct, and we are grateful for their clear example. Our mediation analysis can only detect effects due to changes in protein abundance. Effects mediated by activation states of proteins or missense variants would not be detected. If the true mediator is not observed, its effects will not be detected, and it is possible that another protein could be reported as the best mediator. We have added text to clarify these limitations of mediation analysis in the Results (**lines 170-174**) and reiterate these points in the Discussion (**lines 353-362**).

We wish to note that with mass-spectrometry-based proteomics, polymorphic forms of a peptide cannot be quantified together. We typically identify the more common or reference form of the peptide. For genotypes that produce an alternate form of the peptide, abundance estimates can be reduced or may result in non-detection. We have observed that this can bias quantification and can even lead to "false" local pQTL signals. In this work, we filtered out peptides with known missense mutations and confirmed that this strategy improves quantification. We feel that more work is needed to optimize the analysis of variant peptides but that would fall outside the scope of this manuscript. Our filtering strategy is described in the Methods.

Minor comments from Reviewer #1:

Minor comment #1: Line numbers would have been helpful to this reviewer.

We apologize for neglecting to include line numbers. In the process of first putting together a version of the document for bioRxiv and then adapting for submission to Cell Genomics, we forgot to add back line numbers. They are now included.

Minor comment #2: Page 3 top: At 10 Mb, the criterion for calling a pQTL "local" will span multiple genes and therefore could technically sometimes include trans-acting pQTLs that happen to be located close to the given gene. I assume this wide window is based on the mapping resolution in the linkage data (as opposed to GWAS in an outbred population). It would be helpful to state this explicitly, add the width of a typical pQTL in the data for context.

We use a 10Mb local window because we are comparing pQTL between the CC and DO, which have different mapping resolutions – the DO resolution is more precise due to multiple generations of outbreeding. We also needed to allow flexibility for defining that a pQTL co-mapped in the CC and DO. We initially used a window of 2 Mbp, but that resulted in some “distal” pQTL in the CC that were clearly concordant to “local” pQTL in the DO. We acknowledge that our definition of local pQTL could encompass distal pQTL within 10 Mbp of the gene and have added explanation of this issue in the text (**lines 128-132**).

We generated some plots to highlight the differences in resolution between the CC and DO and to justify the 10Mb window for local pQTL. The peak marker positions of pQTL fall into a narrower region around the gene in the DO compared to the CC (**Figure R1a**). We next estimated support intervals for the pQTL as 1.5-LOD drop. The support intervals for DO local pQTL are narrower (**Figure R1b**), with a median interval of 6.0 Mbp in the CC and 1.8 Mbp in the DO.

Minor comment #3: End of intro: "[...] became fixed during the of the CC" - something is missing here.

The word “breeding” was omitted. The text now reads “became fixed during the breeding of the CC” (**lines 32-33**).

Minor comment #4: Page 4 "cannot be ruled if they unobserved" misses an "are".

Typo has been corrected.

Minor comment #5: Page 4 end of main paragraph: ", which revealed similar levels of concordance [...]": It is not clear from the sentence what these levels of concordance are similar to. I suggest rewording this sentence, as it currently obscures a key result from this section (that mediation signals are shared between CC and DO).

We agree that the original text was confusing and have re-worked it (**lines 194-196**). We have also simplified the description of **Figure S1o** (now **Figure S1I**) and emphasize the matching mediation signals for strong distal pQTL between the CC and DO.

Minor comment #6: Page 5 bottom: "PC1" was not defined in the main text before this use.

We now define PC1 at its first use (**line 209**).

Minor comment #7: Figure 2A: y-axis label (and the equivalent information in the legend) should read "gene position". Proteins do not have positions in the genome; the genes that encode them do. In addition, a scale for dot size (i.e., LOD score) would be helpful.

Corrected.

Minor comment #8: Figure 2J: the legend text for the circus plot (i.e. panel J) runs seamlessly into that for panels K & I. Please delineate these panels more clearly.

Corrected.

Minor comment #9: Figure 2 K & I legend: explain what the LOD score of 12 (dashed horizontal line) denotes. Genome wide significance? FDR = 0.05? The same comment applies to all other plots of this type (e.g. those in Figure 4).

The horizontal lines were intended only as a reference line across plot, particularly when the scales differ (**Figure 2j**). We agree that this was confusing, particularly because almost all other figures had a reference line of around 6. We have adjusted the line to 6 in **Figure 2j** to be consistent. The line is very close to the lenient pQTL threshold for both CC and DO, although we actually identified pQTL based on FDR-based thresholds specific to each population and the level of missingness of the protein (**Methods**).

Minor comment #10: Related, why are the lines in the LOD profiles in Figure 4 (and other figures later on) plotted at LOD=6 rather than LOD=12 like in Figure 2?

See response to above comment. We now consistently use LOD score = 6 for the reference line.

Minor comment #11: Figure 4 legend: "observance of the homozygous PWK genotype": "observance" seems an odd choice of words here. Maybe "observation" or "presence" is better? Also, in "distally control other members" is should read "controls". Finally, panel "h" is incorrectly called "e" in the legend.

Corrections made as suggested. We have adjusted **Figure 4** (and other figures) and shortened the legend.

Minor comment #12: Figure 6k legend: "The relationship is negative" should be deleted.

We agree that this text was unhelpful. It has been removed.

We thank Reviewer #1 for taking time to review this paper and provide constructive comments, which have been used to improve the quality of the manuscript.

Reviewer #2 summary:

The authors have acquired and analyzed liver proteomics data for a fairly large population—116 individuals—and diverse inbred mouse population and they have performed a meta-analysis of these results against the results of a conceptually similar previous 2016 study in a distinct (though related) mouse population of about 220 individuals. The primary difference between the two studies is that this study is entirely on inbred strains, but more or less it is the same idea and makes an ideal case study for population meta-analysis of proteomics data, which to my knowledge has not been done before. This study thus provides a useful baseline for proteomic variation. I have a number of concerns, but all of which should be relatively easily either rebutted, or addressed in a minor revision.

Thanks. We appreciate Reviewer #2's positive assessment of our work.

Specific comment #1: The authors mention in Fig 2D how many local and distal pQTL they identify, with around 25% higher QTL detections in the DO population for the 4556 proteins quantified in both. The DO population is both larger in sample terms and more genetically variable than the CC so this is not especially surprising in one way, but the lower observed heritability would flip in the other direction and favor CCs. What's the relative contribution of sample size, variability, etc?

Reviewer #2 asks an insightful question about the relative contributors to variation between the CC and DO and how they impact heritability in each population as well as the power to map pQTL. There are challenges to doing this rigorously in the case of this study, really for multiple reasons.

Even comparing the levels of genetic diversity between the CC and DO is complicated. The DO have more possible genetic states at each locus, there are 36 potential genetic states in the DO compared to 8 in the CC (ignoring residual heterozygosity). How this contributes to the power to detect genetic effects depends on whether the effects are additive or dominant. The CC sample size is smaller (in terms of unique genomes) which should lead to fewer pQTL mapped. However, this is somewhat mitigated because the inbred CC maximizes the allelic contrasts being compared (2 copies of an allele vs 0 copies) whereas a DO sample will contain a majority of animals with one copy of the allele. The greater heritability in the CC also reflects the sampling of two individuals from each strain. Another factor is that the CC were measured in a more recent TMT mass-spec experiment that included a pooled bridge sample that improves quantification, decreasing noise and thereby increasing heritability and improving power to map pQTL in the CC in comparison to the DO (and founder animals). The trends (CC > Founders > DO) in Figure 1C suggest that both genetic and technical factors are important. Teasing these factors apart is challenging or even impossible with the data in hand. We have added text (**lines 106-110**) to draw attention to some of the challenges in interpreting heritability.

Specific comment #2: I like how you have added the lenient detection comparison (Fig S1F). One related question: for those ~14% of QTLs that are dramatically different between CC and DO, do these fall into any patterns? i.e. genes in large complexes, mitochondrial genes, genes which have fewer sequence variants, ...? I just very quickly checked the top 30 negative correlates in DAVID and did not see any enriched ontologies.

Thanks. We feel that reporting the suggestive findings is important when comparing results across populations, such as the CC and DO. The stringent genome-wide results are most reliable but there are also many false negative results if we strictly adhere to this standard and this results in an underestimation of the reproducibility of pQTL. We also find intriguing biological signals in the lenient results. Examples include the ERCC3, ERCC3, and GTF2H1 co-regulatory network (**Figure S3**), which are remarkably similar between the CC and DO, though only detected at the lenient threshold in both; the same is true for many components of the exosome complex in the CC.

We did not investigate the pQTL with differing effects between the CC and DO largely because we had already identified plenty of stories to follow up. That said, the data are open and represent a rich resource to explore. This is a good example of the type of question that can be posed and investigated further.

In terms of pQTL with essentially flipped effects between the CC and DO, we looked a little deeper for this response. For local pQTL leniently detected in both populations (**Figure S1f**) only three genes had an effects correlation < -0.5. When we broaden this to local pQTL leniently detected in at least one of the populations, it increases to 13 genes, for which we find no enrichment in biological functions based on GSEA. We do not find this particularly surprising because there are so few genes meeting this criterion. Notably, none of these correlations are statistically significant after multiple testing correction. We believe the relative lack of negatively correlated effects is biologically meaningful in terms of the conservation of local genetic effects within a tissue between the CC and DO. Less strong conservation of effects has been observed across tissues within the CC (<https://doi.org/10.1371/journal.pgen.1008537>), where tissue-specific effects can drive differences.

We did grab the two genes with local pQTL with the most negatively correlated effects from the sets shown in **Figures S1f** and **S1g**, respectively. For LGALS9 (**Figure R2** left), which was leniently detected in both populations (counted in both **Figures S1f** and **S1g**), the flip in effects is between the haplotypes from wild-derived strains (CAST, PWK, and WSB) and traditional lab strains. A similar pattern is seen with ACAT2 (**Figure R2** right; most distinctly with PWK), although the pattern is noisier and the pQTL not being leniently detected in the CC (counted in **Figure S1g** but not **S1f**). These patterns are rare in our data and thus we have not

included them in the manuscript text, they could reflect unique examples where wild-derived haplotype effects behave differently between inbred and outbred backgrounds.

Specific comment #3: The authors do mention this, e.g. "COPS8 is the strongest mediator for ..." (page 4) and mention that it "may be less accurately measured in [their] DO sample population". Why would that be the case? I know the LC-MS setup is different (question on that later), but how different is this? E.g. how many peptides are measured for COPS8? How many NAs/0s does its average peptide have? Are QTLs—both cis and trans—that disappear in CC and appear in DO or vice versa, are these more likely to be protein measurements that are the synthesis of only one or two peptides?

In this case, we had a strong expectation to see a local COPS8 pQTL in the DO based on both COPS6 and COPS7A have distal pQTL that map to the COPS8 locus. In the CC, COPS8 has a strong local pQTL and haplotype effects that match the distal pQTL for COPS6 and COPS7A, consistent with COPS8 being detected as a mediator. We see highly consistent effects for the COPS8 local pQTL in the CC (**Figure R3a**) and the distal pQTL for COPS6 and COPS7A near the COPS8 locus in the CC and DO (**Figure R3b**), but COPS8 does not have local pQTL in the DO, even leniently detected. We looked at another related protein, COPS7B, within 4 Mbp of COPS8, but it also does not have a local pQTL in the CC or DO. COPS8 seems to be the best candidate driver in the CC and DO, but we see no obvious explanation for the failure to detect a local QTL in the DO.

Motivated by the Reviewer #2's question, we took a deeper dive into the peptide data to see if they offer any hints as to why COPS8 is inconsistent in the DO. In the CC, we have nine peptides, with six also observed in the DO. One peptide only observed in the CC is not included here due to high levels of missingness. Notably, haplotype effects based on the local region of *Cops8* are evident in the CC peptide data (**Figure R3c**), including the peptides also observed in the DO, suggesting that discrepancy for COPS8 cannot be explained by differing peptides between the CC and DO. We next ran QTL scans for the six peptides observed in the DO, and none possessed obvious QTL signal in the region (**Figure R3d**), consistent with the aggregate summary for COPS8 having no local pQTL.

Examination of the peptide data did not explain the missing COPS8 local pQTL in the DO. We have added text to clarify that we see no obvious explanation (**lines 184-186**).

Specific comment #4: Page 5: "GSEA revealed that proteins related to ribosomes, translation, ... were more abundant in male livers, while ... were more abundant in female livers". Is this expected? This seems surprising to me and I don't think I've ever seen that in literature. Why would ribosomes, translation, etc, be higher in male livers? Certainly not impossible, but I haven't come across it and there is no reference cited, so if the authors are confident in this data, I think they should visualize it in a figure. That the directionality is the same in the DO and CC is certainly a very good clue that it is a real effect, so I don't doubt it, I would just like to see it as it is novel as far as I know and nothing especially relevant came up in a cursory literature search.

As an example, we have included a plot of the sex effects (with 95% CI) for the gene ontology term "ribosome" in this response (**Figure R4**). The sex effects are broadly consistent across populations. The effects on the cytoplasmic ribosomal components are more distinct than the mitochondrial ribosomal proteins (MRP), which the founder data may be under-powered to detect. There is also some consistency in the few proteins with greater abundance in females, for example, EIF2AK4. For higher level gene ontology categories (e.g., translation and peptide biosynthetic process), similar consistency across populations is observed, although more proteins with significantly higher abundance in females are pulled in. Nevertheless, overall, more proteins have greater abundance in males, as expected from the GSEA.

We agree that this is an interesting finding and, while we do not yet understand what is driving it, we wanted to mention it in the manuscript. As demonstrated above, it occurs across the founder strains, CC, and DO, and we have seen (though did not report) a similar pattern of sex-specific differences in ribosomal gene expression in aging mice (<https://doi.org/10.1101/2020.08.28.272260>). The enrichment in various catabolic pathways in

females is less surprising in light of well-known differences between sexes in liver metabolism (e.g., <https://doi.org/10.1371/journal.pone.0242665>). We have added text and references to support the finding (lines 115-119). We had also missed that Romanov et al (<https://doi.org/10.1016/j.cell.2019.03.015>), which we cite extensively in the manuscript, observed this as well in an independent analysis of the same DO sample, which we also now cite at this point in the text. Given page limit constraints and the requested reductions in figures, we did not include a figure.

Specific comment #5: Page 6: "Notably, complex-heritability is consistently higher in the CC than DO and uncorrelated with complex-heritability in the DO". First off, could you directly put the "CC" and "DO labels directly on Figure 3 and Fig S4A/B so readers don't have to go to the legend to see which is red and which is blue? (I know you do it consistently across figures—thanks for that—and you almost always list CC and DO in the figure panel itself, so just a small request here for people like me who quickly forget the color coding).

Second off: This shift in h^2 for complexes between DO and CC looks like exactly the same value as the shift between non-complexes between DO and CC, so couldn't you take any subset of genes and make the same statement? It seems like it's just a uniform downward shift in heritability for all genes in DO compared to CC. The decrease in heritability in complex members compared to non-complex members is very strong and robust, so I have no issue with the overall message. The decrease in heritability is also expected; complex proteins also tend to have far fewer cis-pQTLs compared to an average gene, presumably because PPIs are disproportionately important to regulating their expression. I have no real issue with the text here, it's just not particularly novel by this point.

We have added keys indicating CC and DO colors to both figures.

Possibly Reviewer #2 is referring to the downward shift in the heritability of individual complex members (Figure S4a-b)? We agree that the downward shift in comparing CC and DO is entirely consistent with the shift seen across all proteins. We also agree that the fact that complex members are less heritable and map fewer pQTL is consistent with conventional knowledge of protein complexes and PPI, but we felt our analyses and findings demonstrated it in a unique and clear way.

Our comment on page 6 was not meant in reference to individual proteins (Figure S4a-b) – complex h^2 refers to the principal component summaries of complexes (Figure 3d). Complex h^2 is uncorrelated between the CC and DO, in contrast to the correlated h^2 observed for individual proteins. The CC data contain strain pairs, which could drive higher h^2 for complexes, but it is not clear if the PC-determined weighted average would always work out that way. The intent here was to provide a broad description of Figure 3 and highlight that complex cohesiveness is correlated between CC and DO, complex h^2 is not, but complex cohesiveness and complex h^2 are correlated in the CC (Figure S4e). This does suggest that the strain replicates in the CC are capturing genetic effects that are missed in the DO. We have adjusted the text to improve clarity of this section on protein complexes (lines 212-222).

Specific comment #6: Related to the previous two points: "Protein-complexes previously shown to be driven by sex, such as eIF2B were confirmed...". The authors mentioned in the previous section that ribosomes were more abundant in males, but I don't see a sex effect here; mitochondrial ribosome large and small subunits seem to have no sex effect, and the ER ribosome is not highlighted. I know the previous section was about GSEA and this is about heritability and correlation—but shouldn't it show up in both?

Reviewer #2's comment motivated us to examine the complex PC1 summaries more closely for the ribosomal complexes. We provide a clear example where the complex sex effect size was large for both the CC and DO: eIF2B (Figure R5). For this complex, all proteins have higher abundance in males and the effects are of a similar magnitude. The complex PC1 is effectively a contrast between the sexes, and the large effect size is highlighted in Figure 3. Even in this clear example, sex is more cleanly highlighted in the DO than the CC, where strain also contributes to PC1 variation.

The cytoplasmic ribosomal small subunit has consistently high abundance in males across the three populations (**Figure R6**). For this complex, PC2 contrasts the sexes (**Figure R6b**) whereas PC1 reflects the strain average expression. For the DO, PC1 contrasts the sexes, resulting in a high complex sex effect size (46.0%). In both populations there is a sex effect, but it is more pronounced in the DO.

The mitochondrial ribosomal small subunit, also shown in **Figure S7**, has less striking sex effects than its cytoplasmic counterpart. The effects are most clear in the CC and less so in the DO and founder strains. Consistent with our findings in the manuscript, the abundances of member proteins are highly specific to strain, and the PC1 for CC reflect a combination of strain and sex. Sex does not appear to be a large contributor to PC1 or PC2 for the DO.

We have added labels for the cytoplasmic ribosomal small and large subunits to Figure 3, which highlights the large contribution of sex for the DO for the small subunit. We have also adjusted the text to emphasize that these complex heritability and sex effect size are based on PC1 (**lines 210-211**). This is likely the reason that complex heritability is on average larger in the CC due to capturing the similarity between strain replicates (and potentially reflecting multi-locus effects) whereas in the DO there are no replicates and sex becomes more frequently the driver on the largest dimension of variation.

Specific comment #7: The proteasome difference is fairly impressive: that's quite a few subunits with very clear changes. Are there any known phenotypes?

The inbred CC strains are relatively uncharacterized compared to other mouse strains that have been studied far longer (often >50 years). One of the intended aims of our descriptive analysis of the CC is to motivate researchers to look more closely – could there be, for example, a difference in immune response, or lifespan? We do not know. We added text to highlight this (**lines 336-338**).

Specific comment #8: The mention of "notable exception is Auh". It's interesting and a nice hit, but I'm not sure it's a "notable exception". I found one paper referencing it "associating with" the mitochondrial ribosome, but I wouldn't really call that a complex member in the normal sense of the word. Otherwise, I don't see Auh considered a canonical component of the mitochondrial ribosome in literature, it's not in CORUM as a mito ribosome component, and it is also not really picked up by BioPlex nor STRING. STRING even puts it with cholesterol biosynthesis (although I am always a bit skeptical of STRING). I also don't see AUH on GO's list of "mitochondrial translation". Similar question: why is RPS15 included in this figure? Also some other MRPS genes are measured, but are missing, e.g. MRPS27 and 28 are referenced in the supplemental tables as measured in the CC, but they're not here. PPME1, same question. METTL17 makes sense and fits in both better here in the data, and in BioPlex although not STRING.

We use annotations from Ori et al (<https://doi.org/10.1186/s13059-016-0912-5>), which were manually curated from CORUM and COMPLEAT as well as literature evidence. These annotations were also used by Romanov et al (<https://doi.org/10.1016/j.cell.2019.03.015>). Categorically defining complex membership is a challenging task given that a protein's relationship to a complex may not be black and white. It goes beyond the scope of this work to further manually curate annotations, as none are likely to be perfect. Instead, a strength of our work is that by looking in the data, we can see how tightly co-regulated members appear to be. The converse of AUH is DIS3L and ETF1 which we are able to associate with the exosome even though they were not annotated as members. We have adjusted the text to emphasize that AUH is not a canonical component, which these data support (**lines 292-295**)

We double checked and RPS15 are PPME1 both annotated (in Ori et al) with the mitochondrial ribosomal small subunit (MRSS). Coincidentally, our data suggest that neither RPS15 nor PPME1 are strongly co-regulated with the core MRSS, which we now emphasize in the text. Alternatively, neither MRPS27 nor MRPS28 are included in the annotations, though they clearly correlate with the core complex. We have chosen to not adjust the annotations because it would further complicate the methods, but we describe them in the

text (lines 295-298). We note that the data and analyses like pQTL/mediation and cohesiveness could be used to fine-tune annotations.

Specific comment #9: 6046 strain-protein outliers for 4323 proteins across 58 CC strains sounds like far too many to be accounted for by private spontaneous mutations. Shifman 2006 (PMID 17105354) only detected a few dozen SNPs in far older RI strain populations. I didn't immediately find a number from the 2019 Shorter paper but in any case it should be quite rare. I would expect the vast majority of those 4323 outlier proteins to be noise brought on by measurement issues or normalization issues. How many of those outliers are related to proteins that are measured by a single peptide, for instance? The examples shown are convincing and interesting, but they map to known private variants. The authors do mention this ("measurement error in proteomics is likely from [among other things] the number of peptides used to summarize it...").

We agree – and we did not intend imply that all of these are due to private mutations. Our definition of strain-protein outlier was intentionally lenient, in order to cast a wide net and identify potentially interesting characteristics of many CC strains. The set of 6046 strain-protein outliers also represent the strain-specific groups of functionally related outliers (e.g., in CC013 and CC007), most of which do not have a clear strain-private variant driver although they are clearly biologically real. We have modified wording to clarify that we do not know what causes most of these. Many may be inconsequential but there is an opportunity here to identify interesting CC strains for further study. We have revised text in the Discussion to clarify these points (lines 390-399).

Specific comment #10: The authors mention later that CC013 has a unique liver phenotype and show it in Figure 7—that's neat! But then they next mention "altered complex I function in C007" but do not show or reference it. Did I miss that somewhere? I don't see anything mentioned on its strain page (<https://www.jax.org/strain/029625>).

CC007 was mentioned in the very last sentence of strain section of the Results and is shown in **Figure S7d**. We have moved it to **Figure 7g** to move it out of the supplement and have revised the text to better emphasize it (lines 334-336). As noted above, the CC strains are not well characterized. To the best of our knowledge this is an original observation, and CC007 is a potentially interesting strain to follow up.

Specific comment #11: Methods: The new CC data were acquired on an Orbitrap Fusion Lumos using a different column with different parameters (beads, etc) compared to the DO data from 2016, acquired on an Orbitrap Fusion Tribrid. Also I may have missed it, but I don't see how long the acquisitions were in this study (36 hours per 10-plex in DO).

The runs were shortened from 3 hr to 2.5 hr due to better instrumentation as described in the methods. We have added a sentence to the methods to provide greater detail (lines 645-647).

I'm kind of surprised that the Lumos does not detect more proteins than the Tribrid (both are around 6800). How many individual unique peptides were measured in the two?

There were 118,291 unique peptides in the CC dataset and 134,795 in the DO dataset. While there are more unique peptides detected, the number of plexes was much greater (21 vs 12), and the elution gradients were longer for the DO mouse dataset. The total number of proteins detected in total (8,584 for the CC and 8,687 for the DO) and in more than half the samples (6779 for the CC and 6588 for the DO) were about the same. Overall, shorter gradients and fewer plexes still quantified similar numbers of proteins across the two liver datasets.

Are the measurements more stable in the CC dataset? Could this account for the improvement in observed heritability (Figure 1C) in the CC compared to the founders?

The instrumentation probably had some influence as Lumos is more sensitive

While generally I think the authors' findings match what I would expect, I am worried slightly that some of the meta-analysis is an "oranges to clementines" comparison. Of course I am glad the authors used the newer machine to acquire this data, but it makes it a bit trickier to be certain that differences between CC and DO are due to the population compared to the acquisition. There are probably many ways to do approximations of the effect size of this potential issue. Ideally it would have been great to re-run some of the exact same peptide samples from the DO study that had been sitting in a freezer for 4 years, but in lieu of that, there are a few other ways to estimate differences in data quality. I have thought about this for a few minutes and I always come up with caveats. My general expectation would be that the ground truth of observed heritability for CC vs CC Founders should be equivalent. The increased in observed heritability for CC makes me wonder if it's due to a newer machine, or it could just be the increased number of strains raises the observed heritability (I guess this latter hypothesis is easily testable by bootstrapping, or may indeed be already known in literature as it sounds like a straightforward idea to check).

See response to Reviewer #2's specific comment #1 on the factors contributing pQTL mapping power differences between the CC and DO. Heritability is as much a function of the measurement as genetics.

Consistent with the reviewer's comment, we note that the distribution of heritability between the CC and founder strains are more similar (with a higher median heritability than the DO). This is likely due to the presence of strain replicates, which can increase heritability by reducing measurement error. The CC still have greater heritability estimates overall compared to the founder strains, which could stem from both the precision of the protein measurements (e.g., machine and use of bridge sample) as well as the greater sample size in the CC (116 animals vs. 32). It is difficult to tease these factors apart. We have added some caveats about the different technologies and their impact on interpretation of heritability to the Results text (**lines 104-108**), while also emphasizing that heritability is significantly correlated across populations (**lines 108-110**). In the Methods, we also provide more detail on the differences in heritability estimation among the populations (**lines 773-776**).

Non-scientific comments from Reviewer #2:

Non-scientific comment #1: Typo on page 4: "ruled out if they unobserved" (missing "are") and "BCKDHB for Bckdha and Ppm1k for," (Ppm1k for what?).

We have re-written some of the text in this section, which has fixed these typos. Thank you.

Non-scientific comment #2: Typo page 11: "exosome with a single large effect pQTL local".

This section has also been re-worked in the Discussion, including this sentence.

Non-scientific comment #3: The figure legends are long. Like VERY long. It is four full pages of figure legends for the main story, versus eight pages for the entire results section. I like how the results were concise and to the point for such a complex paper, but the figure legends need to be trimmed. There is a lot more explanation in the legends than needed.

We have extensively revised the figure captions – see comments from Reviewer #3 – however to be self-contained, we needed to include a lot of details. Interpretations have been relocated to the main text where appropriate.

Non-scientific comment #4: I downloaded the full 10 GB dataset and found it surprisingly easy to load the data and work with it. I didn't do much with it besides check that I can see it, but at least everything seems to work. The final output data in phenotypes/debatched is a little unintuitive for someone without a basic bioinformatics background to work with. Maybe this is a good impetus for them to learn how to use it, but it'd be nice if it was in a format where anyone could immediately see "does gene X correlate with gene Y in their

data". Right now they'd have to do a bit of parsing and quite a bit of lookups to convert proteinIDs to gene symbols. Maybe this is fine, as if someone can't do this then it would be difficult to really analyze the data properly. For systems biologists it is certainly usable in its current state, but I think it is still slightly beyond the basic Excel skills of your average molecular biologist.

We have exported Excel files for the protein data from the CC, DO, and founder strains in a wide format (individuals as rows, proteins as columns). The unique protein identifiers are ENSEMBL protein IDs (as some genes have multiple proteins in the data), as such, companion protein annotation tables for each data set have also been exported, providing information such as gene ID, symbol, and genomic position. These will be attached as supplemental tables to the paper, thus allowing more convenient access to them, as Reviewer #2 requests.

Final comments from Reviewer #2:

The authors have generated a wealth of liver proteomics data in 116 individuals from 58 collaborative cross mouse lines and have performed a detailed meta-analysis of this study compared to their earlier and conceptually-similar 2016 work. These results add to the growing, but still quite small, body of literature detailing how large-scale proteomics data can be used to study genetic interactions, and it highlights both a few broad patterns in proteome regulation as well as identifying a select few interesting specific hypothesis either for future examination, or for which they have done their own pilot study here, as for CC013. I have some concerns that the broad analyses comparing DO to CC may be a "clementines to oranges" type comparison due to differences in the acquisition technology which are not addressed: proteomics is changing fast enough still such that a dataset generated 4 years later on the next-gen of LC-MS might well bias the data to be improved in the new dataset regardless of the ground truth. In many cases, the meta-analysis results are so clear-cut that this issue cannot possibly affect the results (i.e. very high cis-pQTL overlap, very low trans-pQTL overlap) but in other cases it is potentially an issue, such as the shift in observed heritability for DO vs CC: which is certainly what one would expect, but may be further biased by the analysis. I have two much more minor concerns, one about the way that complexes were selected (especially for the mitochondrial ribosome) and about making the data trivial for someone to analyze on their own (rather than its current "reasonably easy" state). Converting the *_proteins_data.csv files to an easily-used format would take only maybe a couple hours for someone with basic bioinformatics background to do, but it is certainly not possible in Excel, which unfortunately remains a mainstay, especially for molecular biologists who may be interested in the dataset, but only if it is utterly trivial to analyze. For systems biologists, it is quite fine as it is. Generally, it would be nice to not have to download the entire 10 GB package to just get at the peptide and protein data, even though I enormously appreciate the good organization, reasonably clean code, and completeness of their zip file (e.g. if it does not exceed size regulations, put it as a supplemental table).

I have written an extremely long review because I think this paper is likely to become an excellent reference for the next 5 years going forward as proteomics datasets become increasingly mainstream, and as analysis (and meta-analysis) of proteomics datasets is becoming an increasingly-realistic possibility for people who are not proteomics specialists themselves.

One concern: the take-home message is probably not so clear for those who don't do mouse/population genetics or proteomics. There are several take-home messages that jump to mind; for me I think it's perhaps more interesting (and welcome) that the significant findings from proteomics are so congruent between two distinct and independent populations. That is: significant findings that people pick out from routine proteomics are also likely to be reliable. The abstract seems like it could use some more force to it that will excite an average molecular biologist to say "hey, we should use this dataset" or "hey, we should use proteomics". Right now, I think it only says that point to molecular biologists who are studying large complexes.

We have revised the abstract/summary. All other comments summarized here have been addressed in the responses above.

We thank Reviewer #2 for their thorough read of the manuscript and detailed, constructive comments that have helped improve the quality of the final paper. We note that many of their comments highlight how these data can be further explored.

Reviewer #3 summary:

The MS has been submitted to Cell Genomics, a brand-new journal, so it is difficult to assess its specific suitability but it seems to be within the journal's remit. Assessed on its own terms, the study represents a massive amount of work and is scientifically rigorous, with some interesting findings. My overall impression is that rather too much material has been compressed into a single paper. I assume the study has been submitted as a full paper, where the maximum length of the full text is 45000 characters, which is about 15-16 single spaced pages and roughly the length of the MS main text as submitted (I can't count the number of characters in a PDF). Despite being about maximum length, there is still a great deal of material in the supplement or methods which is barely touched on in the main text. Some of the figures in the main text (again at the maximum) contain far too many sub-figures (eg Fig 6 has 20 sub-figures). 5/7 of the main figures are concerned with protein complexes, but only about half the results text. I don't think this makes for a well-balanced paper.

We went through the entire text and made selective reductions to shorten and focus the presentation. We have also reduced the number of subfigures as well as reorganized and reduced information in the figure captions (See response to non-scientific comment #3 from Reviewer #2). Reviewer #3 makes particular note of Figure 6 which has been reduced and focused on constitutive/inducible proteasome findings.

Note: we broke Reviewer #3's initial statement up into broad comments.

Broad comment #1: The main scientific result is that the genetic architecture of the mouse proteome is largely conserved in two populations - one inbred (the CC: data new to this study) and one outbred (the DO: data from a previous study [Chick et al by the same lab]) - both descended from the same inbred founders, and therefore with the same pool of segregating variation. The study shows that cis-acting pQTLs mostly replicate but trans-pQTL do not, but I think this is likely because the small sample sizes in the study do not have enough power to map trans effects reliably. This finding, whilst not surprising, is important and is well demonstrated, although I have some comments below regarding the analysis performed, particularly regarding non-additive genetic effects. It might be worth contrasting the result with that found in *Drosophila* by Trudy Mackay's group, where very different genetic architectures (but for non-proteome traits) were found in two populations related in a similar way.

Yes, distal pQTL effects tend to be smaller and insufficient power is certainly a contributing factor to the low reproducibility. We commented on this in the revised text in the Results (**lines 162-164**) and Discussion (**lines 347-348**).

We include a reference to work from the Mackay lab on the genetics of aggressive behavior (<https://doi.org/10.1073/pnas.1510104112>) (**lines 351-352**). There are some key differences between the mouse and fly populations that limit extrapolating too greatly (*e.g.*, sample sizes, behavioral phenotypes vs. protein abundance, and greater genetic diversity among the founding mouse strains), but the work does highlight similar differences that can arise when comparing inbred and outbred populations. In order to keep our focus on proteins, which have unique properties like complexes that are regulated through stoichiometric balance), we have not gone into great detail in the text about Shorter et al.

Broad comment #2: My main criticism is that apart from this discovery, the rest of the paper on the protein complexes is more of a descriptive study, albeit often interesting, but without any particular hypotheses or questions to drive the analysis, and therefore rather vague conclusions. This is reflected in the abstract which is also rather tentative. The story arc for the protein complexes is a negative one: that with certain interesting exceptions genetics does not drive complex abundance consistently between DO and CC populations, and that instead the cohesiveness of a complex is its key measure, presumably under environmental or temporal control, and with stoichiometric feedback to ensure the relative abundances of a complex's constituent proteins are matched.

It is a descriptive study. However, the conclusions we present shed new light (we hope) on genetic regulation of proteins, and many present testable hypotheses that are worthy of further investigation. We agree that the original abstract was weak and have revised it.

The relationship between cohesiveness and complex heritability is nuanced, and we find it very interesting. Complex cohesiveness alone does not directly assess the role of genetics on how tightly co-regulated a complex is, which we felt was slightly over-stated in Romanov et al. Heritability is a more direct assessment but can be challenging to estimate and compare across populations. In particular, populations with varying levels of relatedness, *i.e.*, the CC sample includes strain pairs whereas the DO are more evenly related to each other, can better relate relatedness to phenotypic variation. Some of the differences in complex heritability between the CC and DO stems from this.

We feel that classifying it as a “negative” finding is harsh; it demonstrates that complex abundance is often not directly regulated by genetics even in cases when the complex is quite cohesive – this draws attention to the importance of post-transcriptional mechanisms, which have been underappreciated due to the focus on transcriptomics in genetic studies.

Once genetics or sex stops being the organising principle, the MS becomes more a study of special cases, each individually very interesting and with its own figure (such as exosomes, Fig 4, and chaperonins, Fig 5, and proteosomes Fig 6, but I am not sure Fig 7 is worth including) but without a clear take home message. Perhaps some of these stories should be expanded into stand-alone papers in more specialist journals? Each of these mini-studies is very impressive and deserves fuller exposition. But I needed to examine the figures at very high magnification and spend a lot of time figuring out what everything meant to appreciate the stories. It did not help that key interpretations were buried in the figure legends rather than in the main text. This is a general point affecting all the figures and should be addressed. Sometimes there are inadequate descriptions of the figures and one has to guess exactly what some of the figures mean.

Our aim for the narrative was to start with simpler effects (*e.g.*, sex and genetic effects on single proteins) that are highly consistent between the CC and DO, and build to the more complicated protein complexes, which are essentially emergent phenotypes that result in very interesting similarities and differences between the CC and DO. The insights that can be made from these highly unique complexes are intrinsically interesting as well as highlight the value of the data (and the CC and DO). We have reduced some of these figures to simplify the message and to make each more self-contained with its own take home message. We also rewrote the Discussion entirely to do a better job of tying up the story arcs. See Comments regarding specific figures for more detail.

Broad comment #3: One other point regarding the protein complexes. The analyses often use PC1 as a one-dimensional summary statistic (effectively an eigengene). This might not necessarily be the most relevant statistic, because PC1 finds the weighted combination of constituent proteins for a complex with the largest possible variance between individuals. This is tuned to identify situations where the constituents lack cohesiveness, for example where the 26S proteosome exists in two forms in different CC lines. Thus genetic mapping using PC1 would potentially reveal loci associated with variation in the complex and similarly for the

heritability. On the other hand the average of the standardised abundances of the constituents would give a measure of the overall abundance of the complex, and this might have different genetic drivers and a different heritability. Did the authors map QTLs for average protein abundance?

We chose the PC1 summary because it would implicitly scale and down-weight noisy or incohesive proteins, which we viewed as an issue due to potentially imperfect complex annotations and the natural heterogeneity in protein complexes. Conversely, as Reviewer #3 notes, this does mean that a member or subset of members with really strong pQTL could be up-weighted and essentially take over PC1. We accounted for this by first filtering out members with local pQTL or strong distal pQTL, as well as regressing out the effect of sex (in the CC and DO) and diet (just DO) (**Methods; lines 948-957**).

We have confirmed based on the PC loadings that the PC1s generally represent weighted averages of individual protein abundances. Reviewer #3 is correct that this is not necessarily the case with PCs and it was good to evaluate these properties. Please see the response to Reviewer #2's specific comment #2, which is related to discussion of the PC1 summary, but in terms of sex effect size.

We did not map QTL for an unweighted average protein abundance summary. A deeper exploration of summarizing complex protein abundance could be interesting but beyond the scope of this paper. Ultimately, no summary will be perfect; indeed, even the simpler cohesiveness summary is not perfect and would fail to distinguish some patterns of complex heterogeneity. We have added text to this effect to the Discussion (**lines 377-389**).

Broad comment #4: Overall this is an impressive study but needs reworking to make it clearer and cleaner, for example by pruning the number of figures. I found it a very difficult paper to review because it resembles an iceberg, where only a fraction of the tremendous effort that has gone into it is accessible without diving in very deeply.

We have extensively revised the text and figures throughout to improve readability. We acknowledge the complexity of the narrative and hope that the revisions will improve the paper's accessibility.

Detailed comments from Reviewer #3:

Detailed comments on genetic architecture:

Detailed comment #1: I suggest simultaneously estimating additive and dominance h^2 in the DO using two GRMs (GCTA lets you do this, for example). This may have bearing on the differences in protein complex heritability discussed on page 6.

In light of the challenge in comparing heritability across populations where aspects of the measurement technology and experimental design are confounded with population genetic structure, as raised by Reviewer #2, we have reduced our description of heritability and added caveats.

The suggestion to use GCTA for additive and dominance heritability is intriguing, and we are looking into exporting CC and DO genotype data formatted to work with GCTA for future studies. Doing so for this present work goes beyond its scope. In particular, there are differences in how the GRMs would be estimated. The current heritability estimates are based on GRM's estimated from an additive model of the founder haplotype dosages, consistent with the QTL analysis. There is some literature on estimating dominance GRMs from haplotypes; however, understanding and comparing these nuances would distract from the focus of the current paper. Dominance heritability would be highly unstable or even impossible to estimate in the CC where the vast majority of loci are homozygous but could be interesting in the DO.

Detailed comment #2: In Fig1d-f, please add the linear regression lines of (say) CC heritability on DO heritability etc

These figures have been omitted in order to reduce figures and text.

Detailed comment #3: Why not do a simultaneous analysis of CC and DO so that you can test formally if the haplotype effects are the same? That is, fit a model like $y \sim \text{genotype} * \text{population}$, where genotype is either a SNP dosage or haplotype dosage, and population is a factor indicating CC or DO.

This is an interesting idea, but we anticipated some significant challenges that push it beyond the scope of this paper and are possibly exacerbated by these specific data. We used a meta-analysis approach because the CC and DO were collected in independent experiments with differences in the mass-spec technology and study design. The protein measurements themselves are relative quantities, specific to an experiment. We could attempt to normalize them together to mitigate this, but the ability to do that successfully is limited due to the differences in experimental design (*e.g.*, CC have a bridge sample but the DO do not). We played with a few examples and found the test to be seemingly over-powered, *i.e.*, rejecting the null hypothesis of similar haplotype effects due to small differences even when the haplotype effects are concordant based on visual inspection. Similar to our response to Reviewer #3's previous comment, we are interested in joint modeling of the CC and DO, but for the features of these data and take-aways of this study, meta-analysis seems effective.

Detailed comment #4: Define stringent and lenient thresholds in the main text

Thanks for pointing out this oversight. See **lines 123-127** and **lines 134-135**.

Detailed comment #5: In the methods both haplotype- and SN based QTL mapping are described but (I think) only haplotype mapping is used in the results. Is that correct? If so, do the Methods even need to consider SNP-based mapping?

SNP-based association is included in **Figure S2**, overlaid on the haplotype association. As the amount of text in the Methods was fairly brief, we have included it for completeness.

Detailed comment #6: In the Methods, both peptide and protein phenotypes are defined, but only protein phenotypes are reported in the Results, correct?

Yes, that is correct. However, we refer to the peptides in order to explain how the protein abundance estimates are obtained. Our filtration of peptides that contain polymorphisms and overall normalization scheme differs from our previous paper (Chick et al), so we described it for completeness.

Detailed comment #7: Fig 2k,l: What are the bands of pale grey dots on the figures? How do the diagrams under each Manhattan plot relate?

The gray dots represent a mediation LOD score for each of the proteins in the data. The vast majority of proteins are not mediators and thus their mediation LOD score is near the detected pQTL LOD score. Proteins that produce a very low mediation LOD score are candidate mediators of the distal pQTL. The diagrams represent the causal relationships suggested by the QTL and mediation analyses. We have included additional explanation in the figure legends (**lines 464-467**).

Detailed comment #8 (on mediation analysis): Mediation analysis: The Naxd story is unclear and needs more explanation

We have revised the text for clarity (**lines 176-180**).

Detailed comment #9 (on sex effects): Fig 1g,i suggests that DO sex effects are slightly larger on average than CC or founders. Please work out the linear regression lines and test if their slopes are different from 1. (ii) A few CC strains are known to have females and males of similar weight. Do these strains show the same level of

sex difference as strains where the weights are different? In other words, is there a correlation between sex difference in weight and average protein sex difference, across the CC strains?

Following the suggestion to more formally compare sex effects among the populations, we regressed CC sex effects on DO sex effects, CC sex effects on Founder sex effects, and DO sex effects on Founder sex effects (matching **Figure 1d-f**, respectively), while fixing the intercept at 0. The CC had less extreme sex effects than the DO (0.62, 95%CI: 0.61 – 0.63) and founder strains (0.75, 95% CI: 0.73 – 0.76). The DO had more extreme sex effects than the founder strains (1.10, 95% CI: 1.08 – 1.11). The less extreme sex effects in the CC compared to DO matches what we observed for protein complexes.

The dynamics between weight and sex effects, highlighted by CC strains with reduced weight differences based on sex, is beyond the scope of our study, which has not included any analysis of weight data.

Detailed comments on protein complexes:

Detailed comment #10: Define complex cohesiveness and complex heritability in the Results, not just the Methods, and explain their significance.

Thank you for noticing this gap, which we have now corrected; see **lines 199-211**.

Detailed comment #11: I note cohesiveness is defined as the median pairwise correlation of proteins in a complex. What happens with negative correlations? What would happen if it was defined as the minimum absolute correlation? See my comment above relating to PC1

We have not corrected for negative correlations specifically. One can imagine problematic cases such as a complex with primarily two sub-complexes anti-correlated with each other, and then additional uncorrelated individual components, which could result in a cohesiveness around 0. The minimum absolute correlation seems like it would be an unappealing statistic, particularly given that many of these annotated complexes include members that appear peripheral to core members. It also would not represent the average properties of the complex.

Similar to the final paragraph of our response to Reviewer #3's broad comment #3, a deeper examination of complex summaries could be interesting but goes beyond the scope of this work. Cohesiveness as defined produces a fairly consistent summary across the CC and DO, which is appealing.

Detailed comments on figures:

Detailed comment #12: Fig3a,c,e -Presumably the color codes are the same as in other figures (orange=CC, pale blue=DO) . Please indicate on the figure.

Legends now included. Thank you.

Detailed comment #13: Fig 4 (Exosome) d,e,g,h What is the reason for sometimes plotting sex comparisons as scatter plots or as geom_jitter() plots? Surely it would help clarity to use a consistent plot design. A similar point applies to Fig 5g,h

In the jitter plots, the focus is on the outlier strains, which in this case possess the PWK haplotype at *Exosc7* and represent the effects of a pQTL. Splitting the sexes allows us to show how consistent the genetic effect is across the sexes (and across the exosome). In the scatter plots, we are more directly comparing the two sexes and showing the data suggest there is some residual heritability for these complexes after accounting for the detected loci because there is still a significant correlation between males and females that do not possess the large effect haplotypes.

Detailed comment #14: Fig 5 (Chaperonin complex). If have understood the genetic architecture correctly, in DO the genetic control of the Chaperonin complex is driven by a trans pQTL for CCT4 on chr5 that is recessive for NOD. In CC mice it is driven by a combination of the same trans pQTL plus a cis pQTL on chr11 due to the PWK allele.

Yes, your understanding is correct. The pQTL on chr 5 is almost certainly acting locally on Cct6a (and then influencing other members stoichiometrically). Cct6a protein was not detected in the CC.

- (i) Fig 5a,b what are the grey dots? Are the p-values for these Manhattan plots well calibrated (as shown by a qq-plot).

See updated figure legend and response to detailed comment #7. They are mediation LOD scores, defined in **Methods**, and should be largely centered around the peak height of the distal pQTL. As such, the assumptions and expectations of a qq-plot do not apply. We use the mediation LOD scores as an empirical null distribution and identify strongly negative outliers as candidate mediators.

- (ii) What is the evidence that the chr5 QTL is recessive? The fig 5c presumably shows the DO hets as shades of gray (this is not explained in the legend but there is a haplotype dosage greyscale below Fig 5a possibly related to this) which might support a recessive effect but it would be nice to see a formal statistical test.

We acknowledge that this was a somewhat informal observation and no longer refer to it as a recessive effect. In this case, we could test for a non-additive effect in the DO as there are adequate numbers of carriers of 2 copies of NOD, 1 copy, and 0 copies (and will do so outside of this manuscript). This comment was largely inspired by the exosome example where the homozygous carriers of the PWK haplotype are essentially unobserved in the DO, and so formal testing is challenging.

- (iii) if 5f, do dashed lines of any color mean LOD <6 in CC?

That is correct. We have updated the legend to define them (**lines 503-504**). Though these fell below the lenient significance thresholds for pQTL, we include them here based on localizing to complex members and/or associations seen in the larger DO sample.

Detailed comment #16: (i) What is the point of Figs 6e,f? They seem to suggest nothing beyond the Manhattan plots 6g-j. Minor point - the length scaling of the chromosomes on 6e,f is different from that in 6g-j so the chromosomes don't align.

We removed **Figure 6e-f** to simplify and focus the figure. The point was to broadly show the pQTL signal for the proteasome members, and in particular, highlight a hotspot of pQTL in the CC on chromosome 1. We removed these because it became clear they were distracting from the story of the interchangeable immunoproteasome components.

- (ii) I don't understand the story being told in these figures. I also don't understand the colors in 6p,q (why is one black, one brown?) And what do Fig 6k, l n, o tell us?

The original figure was about complexes that displayed genetic effects stemming from multiple loci. This has been simplified to focus on just the proteasome, and specifically genetic control of the balance between the constitutive proteasome and the immunoproteasome.

We have swapped **Figure 6n-s** with **Figure S6** to creates more stand-alone stories for **Figures 6** and **S6**. **Figure S6** is now solely focused on the mitochondrial ribosomal small subunit (MRSS). **Figure 6p** (now **Figures S6c**) shows that the abundance of MRPS7, a core protein of the MRSS, is remarkably consistent within CC strain, which suggests genetic regulation. Moreover, these proteins do not map strong pQTL, suggesting that multiple loci of small effects that we are not powered to detect combine to produce the strain-specific abundance pattern. **Figure 6q** (now **Figures S6d**) shows a similar message, but for the MRSS PC1 instead of a single

protein. The complex heritability is a little higher based on summarizing over multiple cohesive proteins. The difference in color was just meant to differentiate them. We have adjusted this, making the MRPS7 plot have solid light red color (CC color) points and the PC1 plot have open light red color points. The CC color is used because the plots represent CC data, which we wanted to indicate visually so readers did not think one was associated with the DO, as often there is a CC version and a DO version to matching figures.

Figures 6k-l show the relationship, driven by genetic effects, between two analogous components from the interchangeable proteins of the proteasome in the CC. Mice with the WSB haplotype at *Psmb9* have more PSMB9 than PSMB6, which is consistent with effects seen in the DO and founder strains. The updated figures are both simpler and self-contained, which we hope will improve reader understanding.

Detailed comment #17: Fig4, Fig5, Fig 6: Some of the plot types are the same between the figures, but differ in small ways. eg. 4h 5h use different coloring rules. It would help to be completely consistent across figures.

We have checked and updated the figures, adjusting some of the color schemes to try to minimize overlap in color scales, though it is challenging when multiple color scales are needed in a figure (*e.g.*, CC/DO, founder strain identity, and correlation) while also avoiding schemes that would be challenging for colorblind readers. Where there are differences, they are noted in the legend and convey a meaning specific to that figure.

For **Figures 4h** (now **4g**) and **5h** specifically, the color schemes differ because they are conveying subtly different systems. **4g** represents a single pQTL, and so color simply corresponds to the haplotype at *Exosc7*. **5h** is more complicated, representing distinct haplotypes effects at two pQTL (distinguished by founder haplotype and shape of point). Coloring the non-carriers of the haplotypes of interest at two pQTL would not make sense because it represents two loci and also make the image harder to understand.

Detailed comment #18 (on Discussion): Not sure I follow this argument:

"comparing the CC to the DO reveals the impact of an inbred genetic background on a number of protein-complexes, due to recessive effects, or conversely, the lack of dominance genome-wide. Furthermore, CC strain replicates can capture multi-locus interactions, i.e., epistasis, by fixing alleles at multiple loci within a strain."

Why is the CC better for dissecting epistasis than an outbred population like the DO?. Also I am not sure why recessive or dominant effects are easier to detect in the CC, if that is what is being claimed. In fact the reverse is surely the case, because inbred populations cannot distinguish additive from non-additive genetic effects - DO heterozygote genotypes are the only sources of information that can make this distinction.

We did not mean to imply that the CC alone could dissect epistasis, but rather that by comparing the CC to the DO or even CC lines to each other, it is possible to see unique systems or patterns that appear to be highly consistent with non-additive genetic effects, such as CC007's unique mitochondrial respiratory chain complex I abundance (**Figure S7d**). Similarly, we agree that the CC cannot distinguish between additive and dominance effects, as they are confounded in the CC, but can be inferred by comparison to the DO. The DO could technically distinguish all these genetic effects, though it may require very large sample sizes to observe enough homozygotes of each founder haplotype and be essentially impossible to capture the multi-locus effects that are fixed and associated with unique protein networks in the CC. We have revised text in the Discussion to improve clarity (**lines 393-397**).

We thank Reviewer #3 for their time and energy in reviewing our work. Their detailed comments motivated us to substantially revise and simplify the figures and text.

Figure R1. Mapping resolution is finer in the DO than the CC. (a) LOD score by distance between pQTL position and the middle of gene for the CC (left) and DO (right). Red dashed lines represent 10 Mbp upstream and downstream from gene middle, which was used to define a pQTL as local (*cis*) or distal (*trans*). LOD score > 6 was used as an *ad hoc* lenient threshold for determining the pQTL included in the plot. (b) Comparison of support intervals (1.5-LOD drop) for pQTL position between the CC and DO (left). Red Identity line included for reference. Zooming in emphasizes that the CC have generally broader support intervals, *i.e.*, reduced mapping resolution, as expected (right).

Figure R2. A small number of proteins have negatively correlated haplotype effects between the CC and DO, such as (left) LGALS9 and (right) ACAT2. Bars represent standard errors. Both cases represent flipped effects relative to the wild-derived founders (CAST, PWK, and WSB).

Figure R3. Lack of COPS8 local pQTL signal present in DO peptide data. (a) *Cops8* has a local pQTL detected in the CC, characterized by a high effect group for AJ (A), B6 (B), and NOD (D), and low effect group in 129 (C), NZO (E), and CAST (F). (b) *Cops7a* and *Cops6* possess distal pQTL in both the CC and DO that co-map with the *Cops8* local pQTL detected in the CC. In the CC, COPS8 abundance mediates both distal pQTL. The founder haplotypes effects for the distal pQTL are similar between the CC and DO. Bars represent standard errors. (c) The peptides for COPS8 have similar effects compared to the local pQTL, though with weakened signal compared to the composite protein estimate. (d) Peptide QTL scans covering the local region for *Cops8* reveal no clear signal for any of the peptides. Vertical lines marks the midpoint of *Cops8*.

Figure R4. Sex effects on proteins associated with the ribosome gene ontology term in the CC, DO, and founder strains. Effects are shown as 95% confidence intervals. Statistically significant non-zero effects are colored. Only proteins analyzed in all three populations are shown.

a**b**

CC complex sex effect size: 39.9%

**c**

DO complex sex effect size: 56.9%

Figure R5. The eIF2B complex is primarily driven by sex in both the CC and DO. (a) Sex effects for the individual proteins of the eIF2B complex, represented as 95% confidence intervals, for the CC, DO, and founder strains. Statistically significant non-zero effects are colored. Only proteins analyzed in all three populations are shown. (b) Comparison of the PC1 and PC2 summarizing the complex in the CC. PC1 largely reflects sex, which explains 39.9% of the variation (effect size). (c) This is even more pronounced in the DO, where sex explains 56.9% of the variation in PC1.

a**b****c**

DO complex sex effect size: 46.0%

Figure R6. Members of the cytoplasmic ribosomal small subunit are more abundant in males in the CC, DO, and founder strains, consistent with Figure R4. (a) Sex effects for the individual proteins of the cytoplasmic ribosomal small subunit, represented as 95% confidence intervals, for the CC, DO, and founder strains. Statistically significant non-zero effects are colored. Only proteins analyzed in all three populations are shown. (b) Comparison of the PC1 and PC2 summarizing the complex in the CC, colored by (left) strain and (right) sex. PC1 largely reflects strain instead of sex, which only explains 0.8% of the variation (effect size). In the CC, PC2 instead largely absorbs sex. (c) In the DO, PC1 still largely reflects sex, where it explains 46.0% of the variation.

Figure R7. Members of the mitochondrial ribosomal small subunit are more abundant in males in the CC, which is less pronounced in the DO and founder strains. (a) Sex effects for the individual proteins of the mitochondrial ribosomal small subunit, represented as 95% confidence intervals, for the CC, DO, and founder strains. Statistically significant non-zero effects are colored. Only proteins analyzed in all three populations are shown. (b) Comparison of the PC1 and PC2 summarizing the complex in the CC, colored by (left) strain and (right) sex. PC1 reflects a combination of strain and sex, resulting in sex explaining 6.3% of the variation (effect size). In the CC, PC2 also appears to be a combination of strain and sex. (c) In the DO, neither PC1 nor PC2 clearly represent sex, which accordingly explains 2.9% of the variation in PC1.

Second Review, Referee Reports:

Referee #1

Thanks to the authors for addressing my comments during revision (I was reviewer #1). My remaining comments are about the presentation and flow of the manuscript:

1. Section starting line 166: I find it odd that most of the space in a section titled "Mediators of strong distal genetic effects are concordant between the CC and DO." is devoted to the few discordant exceptions and does never really emphasize high concordance. In this way, the text does not really support the headline. It may be enough to discuss a single discordant mediator here. Alternatively, if the point of this section is to highlight shortcomings of mediation (as the authors do in the Discussion, and which I agree is important), it may be better to change the header of this section.

2. Figure 2 j & l: These (and all other similar) panels remain confusing. This is because the reader is asked to process two sets of information with opposite directions of importance. In one trace (the LOD profile), "higher is better". In the other (the mediation LOD points), "lower is better". As a result it took me a long time to understand why NAXD is not the most significant mediator in the CC panel of 2l (after all, it seems to sit right at the top of a peak!), and why it is supposed to be the most significant mediator in the DO panel (it is much lower than TUBGCP3). This is independent of whether NAXD is biologically a true mediator; my comment is purely about the display of the data. My strong recommendation would be to separate these two kinds of information into separate panels stacked on top of each other. I'd also invert the scale for the mediation LOD such that "higher is better" in both panels.

3. An overall comment on the figures: I agree with my fellow reviewers that the figures contain too much information and devote too much space to what are unimportant details for the key points this paper seeks to make. This is especially true for the enormous Figure 3, the big and mostly redundant multiple heat maps in Figure 5, and the myriad distribution plots in Figure 7. I don't think many readers will study these panels in detail, nor do they need to in order to understand the paper. My advice is to find more compact ways to visually support the key point(s) the authors wish to make and relegate the details to supplement. Currently, there is a real risk that readers will miss the forest for the trees.

Minor comments:

4. In the submission questions, the authors answered "no" to the question "Does your manuscript report custom computer code or introduce a new algorithm?". Given there is custom code in the Figshare, shouldn't this be "yes"? I do understand that new computational tools are not the main focus of the paper; maybe this is the reason for answering "no" here.

5. Line 143: "that" is repeated

Referee #2

Reviewer comments for the revised Cell Genomics article "Regulation of protein abundance in genetically diverse mouse populations" by Keele et al.

Revision comments

(1) Re: comment 5: "Possibly Reviewer #2 is referring to the downward shift in the heritability of individual complex members (Figure S4a-b)?" Yes, that's what I was referring to. The explanation is fine, both in the reviewer text and in the updated manuscript.

(2) Re: Several comments on sex-related differences in the ribosomes: Thanks for the additional explanation and reviewer figures. It will be interesting for people to follow this up in future studies, as it has a really quite dramatic apparent effect, and it is something an effect that I would not necessarily expect to see at the transcript level (since so many factors that affect complex expression are unique to mRNA or protein expression).

General: I read through the paper again and I find the highlighted examples very interesting, including the ones that I mentioned which the authors did some revision-specific analysis for. Of course there are always practical limits to the length of a manuscript, so I think it is almost inevitable that such a paper is merely the visible tip of the "iceberg" as reviewer #3 notes. The paper itself is nice, but the biggest value is certainly for those who can do their own deep dive in the data. This was already feasible before due to the very clean code and data, but it is now even easier with the new supplemental table S2. Even someone with the most rudimentary bioinformatics skills (i.e. Excel) can use it.

I think this paper is really rather necessary, especially as it looks like the Churchill and Gygi lab have several other fairly similar datasets coming in the near future, (e.g. in the rebuttal, the authors link a biorxiv paper posted on April 2021, Gyuricza et al., for heart proteomics, from more or less the same authors/labs). This paper lays essential groundwork to be able to take future datasets at face-value without spending 4 figures on cross-study comparisons and other QC checks. I think is also very helpful for the general population-genetics-of-gene-expression community as it is a very concise meta-analysis that shows more or less what data we should expect to see consistent, and inconsistent, across analogous studies. It will be interesting to see how that holds up across tissues and so forth once the authors' labs' other studies are further along.

The authors highlight a number of interesting possibilities for mechanistic papers, but for a journal like "Cell Genomics" this "big picture" viewpoint seems appropriate. (Not that I really know what Cell Genomics' ideal target is since it's a new journal.) Presumably the authors are following up on sex differences in ribosomes, or CC007 livers, or one of the other mechanistic hypotheses that are convincingly developed here. Those experiments would not only take quite a bit of time to add, but they would also rather take this paper off its initial topic, and I think this study stands alone in its current state.

Referee #3

I have reviewed the revised manuscript. Although the authors did not make all the revisions I requested they gave reasoned arguments why they decided not to make all the changes. The paper's readability is improved in the revision, so I think it is suitable for publication.

Response to reviewer comments for the revision of
"Regulation of protein abundance in genetically diverse mouse populations"
[CELL-GENOMICS-D-20-00070]

Reviewer #1 summary:

Reviewer #1: Thanks to the authors for addressing my comments during revision (I was reviewer #1). My remaining comments are about the presentation and flow of the manuscript.

Specific comment #1: Section starting line 166: I find it odd that most of the space in a section titled "Mediators of strong distal genetic effects are concordant between the CC and DO." is devoted to the few discordant exceptions and does never really emphasize high concordance. In this way, the text does not really support the headline. It may be enough to discuss a single discordant mediator here. Alternatively, if the point of this section is to highlight shortcomings of mediation (as the authors do in the Discussion, and which I agree is important), it may be better to change the header of this section.

We agree that the section was long and focused too much on examples of discordance between the CC and DO. We have reduced the length and now only describe in detail the Tubg1 distal pQTL example. We also changed the section to "Mediators of strong distal genetic effects detected in both CC and DO are concordant" to emphasize that mediation concordance is dependent on detecting strong distal pQTL in both populations, which is an important message for investigators who aim to validate findings between the CC and DO.

Specific comment #2: Figure 2 j & l: These (and all other similar) panels remain confusing. This is because the reader is asked to process two sets of information with opposite directions of importance. In one trace (the LOD profile), "higher is better". In the other (the mediation LOD points), "lower is better". As a result, it took me a long time to understand why NAXD is not the most significant mediator in the CC panel of 2l (after all, it seems to sit right at the top of a peak!), and why it is supposed to be the most significant mediator in the DO panel (it is much lower than TUBGCP3). This is independent of whether NAXD is biologically a true mediator; my comment is purely about the display of the data. My strong recommendation would be to separate these two kinds of information into separate panels stacked on top of each other. I'd also invert the scale for the mediation LOD such that "higher is better" in both panels.

The overlaid genetic association and mediation plots convey a lot of information compactly, which can be both good and bad. We have elected to keep them as they are because our goal in almost all cases is to show similarities and differences between the CC and DO. If we were to break them into their individual components, as aligned genome and mediation scans, the figures would need to be expanded even further. For example, **Figures 2J-L** currently show four component plots representing both consistent and inconsistent examples between the CC and DO. This would have to be expanded to eight component plots if they were separated.

A key take-away from these plots is that the best candidate mediators are generally right at the QTL peaks. Inverting the mediation scale while still overlaying the genome and mediation scans is tricky because the numeric scales would no longer be the same. Whereas the current plots

show mediation scores as conditional LOD scores and thus they have the same numeric scale. We do now emphasize in the figure legends that the mediation score is a conditional LOD score, which may help explain the inverted scales. We appreciate Review #1's feedback but ultimately decided to keep the figures as they are for the sake of compactness and efficient visual comparison between CC and DO.

Specific comment #3: An overall comment on the figures: I agree with my fellow reviewers that the figures contain too much information and devote too much space to what are unimportant details for the key points this paper seeks to make. This is especially true for the enormous Figure 3, the big and mostly redundant multiple heat maps in Figure 5, and the myriad distribution plots in Figure 7. I don't think many readers will study these panels in detail, nor do they need to in order to understand the paper. My advice is to find more compact ways to visually support the key point(s) the authors wish to make and relegate the details to supplement. Currently, there is a real risk that readers will miss the forest for the trees.

Our goal was to make highly detailed figures that highlight that the value of these populations as resources, and thus displayed the extent of signal underlying their similarities and differences across a number of examples. For instance, with the proteasome example (**Figure 5**), the consistency in the correlation patterns of the complex across the CC, DO, and founder strains is remarkable. We have made some adjustment to the figures to reduce information load. We adjusted **Figure 3** by reducing the number of complexes that are labeled and increasing the label size and thus emphasis on complexes that are discussed in the text. We moved the chaperonin plots in **Figure 4** to **Figure S6**. We moved the founder-based plots of **Figure 5** to a new **Figure S7** (moving the original **Figure S7** to **Figure S8**). We hope this strikes a balance of reducing the figures and legends in the main text, while also not hiding the rich data that support the Results.

Minor comments from Reviewer #1:

Minor comment #1: In the submission questions, the authors answered "no" to the question "Does your manuscript report custom computer code or introduce a new algorithm?". Given there is custom code in the Figshare, shouldn't this be "yes"? I do understand that new computational tools are not the main focus of the paper; maybe this is the reason for answering "no" here.

We had interpreted this as meaning that the manuscript was focused on the presentation of a new algorithm. We have changed our answer to "yes".

Minor comment #2: Line 143: "that" is repeated

Correction made.

We thank Reviewer #1 for their time and helpful comments which we feel have improved the quality of our presentation.

Reviewer #2 summary:

(1) Re: comment 5: "Possibly Reviewer #2 is referring to the downward shift in the heritability of individual complex members (Figure S4a-b)?" Yes, that's what I was referring to. The explanation is fine, both in the reviewer text and in the updated manuscript.

We are glad our response helped clear up the misunderstanding.

(2) Re: Several comments on sex-related differences in the ribosomes: Thanks for the additional explanation and reviewer figures. It will be interesting for people to follow this up in future studies, as it has a really quite dramatic apparent effect, and it is something an effect that I would not necessarily expect to see at the transcript level (since so many factors that affect complex expression are unique to mRNA or protein expression).

We appreciate Reviewer #2's various questions and suggestions, which allowed us to demonstrate the type of analyses that can be performed with these data, even though we were not able to include the additional findings in our manuscript. This type of exploration highlights the value of these data as a resource. We did not anticipate that male mice would have increased abundance in proteins related to ribosomes and translation, though the consistency across these data (and the other CC/DO data sets we have) is remarkable and needs further study. Many of these data sets also include transcripts, which could also be incorporated, allowing for a more detailed picture of the role of sex on these genes from RNA to protein.

General: I read through the paper again and I find the highlighted examples very interesting, including the ones that I mentioned which the authors did some revision-specific analysis for. Of course, there are always practical limits to the length of a manuscript, so I think it is almost inevitable that such a paper is merely the visible tip of the "iceberg" as reviewer #3 notes. The paper itself is nice, but the biggest value is certainly for those who can do their own deep dive in the data. This was already feasible before due to the very clean code and data, but it is now even easier with the new supplemental table S2. Even someone with the most rudimentary bioinformatics skills (*i.e.*, Excel) can use it.

We agree with Reviewer #2 that the manuscript is something like an "iceberg", particularly in the second half, where we detail some protein complex examples as essentially stories of their genetic regulation. We appreciate that Reviewer #2 finds these stories interesting, valuable for generating hypotheses for future studies, and also notes that the constraints of a manuscript make it unavoidable. We also appreciate Reviewer #2's suggestion to include data as a supplemental table, which should be more convenient for many readers rather than downloading the entire figshare repository.

I think this paper is really rather necessary, especially as it looks like the Churchill and Gygi lab have several other fairly similar datasets coming in the near future, (*e.g.*, in the rebuttal, the authors link a biorxiv paper posted on April 2021, Gyuricza et al., for heart proteomics, from more or less the same authors/labs). This paper lays essential groundwork to be able to take future datasets at face-value without spending 4 figures on cross-study comparisons and other QC checks. I think is also very helpful for the general population-genetics-of-gene-expression community as it is a very concise meta-analysis that shows more or less what data we should expect to see consistent, and inconsistent, across analogous studies. It will be interesting to see

how that holds up across tissues and so forth once the authors' labs' other studies are further along.

We are grateful for Reviewer #2's appreciation for the value of this work, particularly in the context of upcoming studies, and we completely agree.

The authors highlight a number of interesting possibilities for mechanistic papers, but for a journal like "Cell Genomics" this "big picture" viewpoint seems appropriate. (Not that I really know what Cell Genomics' ideal target is since it's a new journal.) Presumably the authors are following up on sex differences in ribosomes, or CC007 livers, or one of the other mechanistic hypotheses that are convincingly developed here. Those experiments would not only take quite a bit of time to add, but they would also rather take this paper off its initial topic, and I think this study stands alone in its current state.

We agree with Reviewer #2's summary of the current manuscript. There are many insights from this paper that can be followed up and further explored. Our goal was to do a "big picture" paper as a rigorous overall comparison of the CC and DO while providing examples of interesting stories that can be further explored in future work. We are grateful that Reviewer #2 values our approach.

We thank Reviewer #2 for all their efforts in reviewing this manuscript. We are particularly grateful for their appreciation of this work (and associated data) as a resource.

Reviewer #3 summary:

I have reviewed the revised manuscript. Although the authors did not make all the revisions I requested, they gave reasoned arguments why they decided not to make all the changes. The paper's readability is improved in the revision, so I think it is suitable for publication.

We are glad that Reviewer #3 accepts the revisions we made as well as our arguments against some of their requests. They had also suggested on simplifying some figures in their initial comments, which we have also further adjusted in these final revision steps (see response to Reviewer #1's specific comment #3).

We thank Reviewer #3 for their input throughout this review process. Their comments have helped us craft a cleaner and more readable manuscript.